# Sleep-dependent reconsolidation after memory destabilization in starlings

Timothy P. Brawn [1,2,3], Howard C. Nusbaum[2] & Daniel Margoliash [1,2]

Reconsolidation theory describes memory formation as an ongoing process that cycles between labile and stable states. Though sleep is critical for the initial consolidation of a memory, there has been little evidence that sleep facilitates reconsolidation. We now demonstrate in two experiments that a sleep-consolidated memory can be destabilized if the memory is reactivated by retrieval. The destabilized memory, which can be impaired if an interference task is encountered after, but not before, the memory is reactivated, is then reconsolidated after sleep. In two additional experiments, we provide evidence suggesting that the learning of the interference task promotes the subsequent sleep-dependent enhancement of the original memory. These results provide novel insight into the complex mechanisms of memory processing, as well as critical evidence supporting the view that long-term memory formation involves a dynamic process of sleep-dependent consolidation, use-dependent destabilization, and sleep-dependent reconsolidation.

---

[1] Department of Organismal Biology and Anatomy, University of Chicago, Chicago, IL 60637, USA. [2] Department of Psychology, University of Chicago, Chicago, IL 60637, USA. [3]Present address: Picower Institute for Learning and Memory, Massachusetts Institute of Technology, Cambridge, MA 02139, USA. Correspondence and requests for materials should be addressed to T.P.B. (email: tpbrawn@mit.edu)

Memory consolidation has traditionally been described as a process in which a newly acquired, labile memory trace is converted into a stable form that is resistant to disruption by subsequent interference from other experiences[1]. According to this perspective, labile memories become increasingly stable until the consolidation process is complete and the trace persists as a long-term memory. This conception has been challenged by observations that memory reactivation (i.e., retrieval) can return a stable memory back to a labile state. Once destabilized, the memory is susceptible to disruption from amnestic manipulations (e.g., protein synthesis inhibitors, interference from other learning, etc.) and needs to be consolidated again. Accordingly, reconsolidation theory posits memory storage after acquisition as a dynamic process that begins with the initial consolidation of the memory but is then followed by cycles of memory destabilization and reconsolidation[2–6].

The study of reconsolidation has borrowed its general methodological approach from the more established field of memory consolidation[3]. A central tenet in many theories of memory consolidation is that brain activity during sleep is critical to the consolidation of newly acquired memories[7–11]. Yet, a role for sleep in reconsolidation has received little attention[12,13]. Does sleep play a similar role in reconsolidating a previously acquired memory that has been destabilized as it does in the initial consolidation of that memory?

We have previously shown that starlings manifest sleep-dependent memory improvements after auditory learning that parallel the general pattern found in sleep consolidation of human auditory learning[14]. Starlings learned to classify novel starling songs and were retested after retention intervals that consisted of wakefulness or sleep. Classification performance decreased non-significantly across waking retention but increased significantly after any retention interval with sleep[15]. Given that human studies often show significant performance decreases across the day[14,16–18], we further examined how sleep affected memories perturbed by interference. Starlings were trained to classify a second pair of novel songs, which produced interference between the two tasks such that performance on both tasks was significantly impaired across waking retention. Nonetheless, sleep consolidated the classification memories such that performance

on both tasks was significantly better after sleep[19], effectively eliminating the amnestic interference between tasks.

We have thus demonstrated that sleep consolidates auditory memories in starlings[15,19] and that these memories can be interfered with by similar learning experiences prior to being consolidated by sleep[19,20]. This establishes the conditions to examine whether sleep-consolidated memories can be destabilized and then reconsolidated by sleep. Here we test whether a destabilized memory is reconsolidated by sleep. In the first two experiments, we demonstrate that post-sleep retrieval can destabilize memories in starlings, allowing for performance impairments if interference is encountered after, but not before, the retrieval. Critically, we show that a night of sleep reconsolidates the memory, resulting in performance recovery and stabilization after sleep. In two more experiments, we show that training on the interference tasks promotes a sleep-dependent enhancement of the primary learning task even without explicit task retrieval, a novel observation indicating that an instance of learning can enhance related memories through a sleep-dependent mechanism.

## Results

**Sleep reconsolidation after retrieval-based destabilization**. The first experiment (Fig. 1) was designed to answer three questions concerning memory consolidation, destabilization, and reconsolidation. First, does a second night of sleep produce a second bout of sleep consolidation for the classification memory when an interfering task is not encountered? That is, does a second night of starling sleep produce memory benefits similar to the first night of sleep? Second, does the retrieval of a sleep-consolidated memory destabilize the memory such that it becomes susceptible to interference again? That is, will learning a new pair of song stimuli in task-B after being tested on task-A produce performance impairments for task-A across waking retention on the second day? Third, if the memory for task-A is destabilized, does sleep then reconsolidate the classification memory such that performance improves or stabilizes after a second night of sleep?

We started by assessing performance accuracy averaged across all conditions on the immediate post-training classification task-A test, which was $69.2 \pm 1.6\%$ (mean ± SEM). This performance level was significantly greater than chance performance of 50%

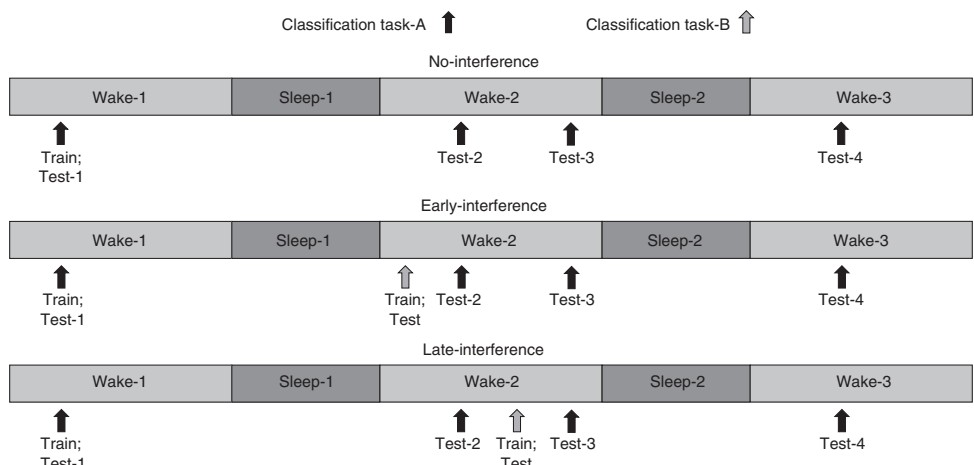

**Fig. 1** Experimental design for experiment-1. Thirty-six starlings were trained to classify 5 s segments of novel starling song. Starlings were trained on classification task-A from 9:30 a.m. to 12:00 p.m. and tested four times thereafter: immediately after training at 12:00 p.m. on day-1 (test-1), at 12:00 p.m. on day-2 (test-2), at 5:30 p.m. on day-2 (test-3), and at 12:00 p.m. on day-3 (test-4). In the interference conditions, starlings also were trained and tested on classification task-B on day-2, either in the morning from 7:30 a.m. to 11:00 a.m. before task-A was tested (early-interference) or in the afternoon from 1:00 p.m. to 4:30 p.m. after task-A was tested (late-interference). Task-B entailed the same Go/No-Go operant training and testing procedure as task-A but utilized a different pair of novel song stimuli than task-A. Each starling completed all three conditions

($t_{35} = 11.65$, $P < 0.0001$, one-sample $t$-test, $n = 36$), confirming that a single training session produced significant auditory classification learning in the starlings[15,19,20]. Likewise, performance accuracy averaged across the two interference conditions on the post-training classification task-B test was 71.9 ± 2.4%, also significantly above chance ($t_{35} = 9.19$, $P < 0.0001$, one-sample $t$-test, $n = 36$). The immediate post-training test performance for each task did not differ across the conditions ($F_{2,70} = 0.04$, $P = 0.96$, repeated-measures ANOVA, $n = 36$ for task-A and $t_{35} = 0.82$, $P = 0.42$, paired $t$-test, $n = 36$ for task-B).

To evaluate performance on task-A following training and retention, we conducted a 3 (condition: no-interference, early-interference, late-interference) × 4 (test: tests 1–4) repeated-measures ANOVA ($n = 36$). While there were no significant effects for condition ($F_{2,70} = 0.06$, $P = 0.94$) or the condition × test interaction ($F_{6,210} = 1.29$, $P = 0.26$), there was a main effect for test ($F_{3,105} = 27.92$, $P < 0.0001$). Task-A classification performance in the no-interference condition (i.e., retention without task-B interference) increased by 5.4 ± 1.5 percentage points from test-1 (12:00 p.m., day-1) to test-2 (12:00 p.m., day-2), demonstrating a significant performance improvement after the first night of sleep ($t_{35} = 3.62$, $P = 0.0009$, paired $t$-test, $n = 36$). Performance remained stable thereafter, showing non-significant increases of 1.2 ± 1.7 percentage points after waking retention at test-3 (5:30 p.m., day-2) ($t_{35} = 0.71$, $P = 0.48$, paired $t$-test, $n = 36$) and 1.1 ± 2.0 percentage points after a second night of sleep at test-4 (12:00 p.m., day-3) ($t_{35} = 0.52$, $P = 0.61$, paired $t$-test, $n = 36$) (Fig. 2a; Supplementary Fig. 1a for additional no-interference data). By comparison, classification performance in the early-interference condition (i.e., task-B interference on day-2 before the task-A test-2) increased significantly after a night of sleep by 5.8 ± 1.5 percentage points from test-1 to test-2, ($t_{35} = 3.85$, $P = 0.0005$, paired $t$-test, $n = 36$), confirming that the memory for task-A was stabilized after sleep and not disrupted by the task-B interference encountered that morning before the task-A retest[19]. Performance remained stable across day-2 with a non-significant reduction of 1.0 ± 1.4 percentage points at test-3 ($t_{35} = 0.69$, $P = 0.49$, paired $t$-test, $n = 36$). However, in contrast to the no-interference condition, classification performance at test-4 showed a significant improvement of 5.3 ± 1.3 percentage points after a second night of sleep ($t_{35} = 3.97$, $P = 0.0003$, paired $t$-test, $n = 36$) (Fig. 2b). Classification performance in the late-interference condition (i.e., task-B interference on day-2 after the task-A test-2) increased significantly after a night of sleep by 5.6 ± 1.5 percentage points from test-1 to test-2 ($t_{35} = 3.69$, $P = 0.0008$, paired $t$-test, $n = 36$). In contrast to the other conditions, performance decreased significantly across waking retention by 3.6 ± 1.5 percentage points at test-3 ($t_{35} = 2.35$, $P = 0.02$, paired $t$-test, $n = 36$). Nonetheless, performance increased significantly by 6.8 ± 1.6 percentage points after a second night of sleep at test-4 ($t_{35} = 4.35$, $P = 0.0001$, paired $t$-test, $n = 36$) (Fig. 2c; Supplementary Fig. 1b for additional late-interference data). This directly demonstrates that the memory for task-A, which had been destabilized by task-A retrieval and impaired by subsequent task-B interference, was reconsolidated by sleep on the second night.

**Cycles of destabilization and sleep reconsolidation.** In experiment-1, we established that retrieval destabilized a sleep-consolidated memory, allowing for subsequent interference to impair task performance across waking retention. Nonetheless, sleep reconsolidated the memory. Here we expand upon experiment-1 by monitoring memory performance on task-A for five consecutive days while the interference conditions also receive daily training on a novel interference task (B, C, and D) over days 2–4 (Fig. 3). Reconsolidation theory proposes that a

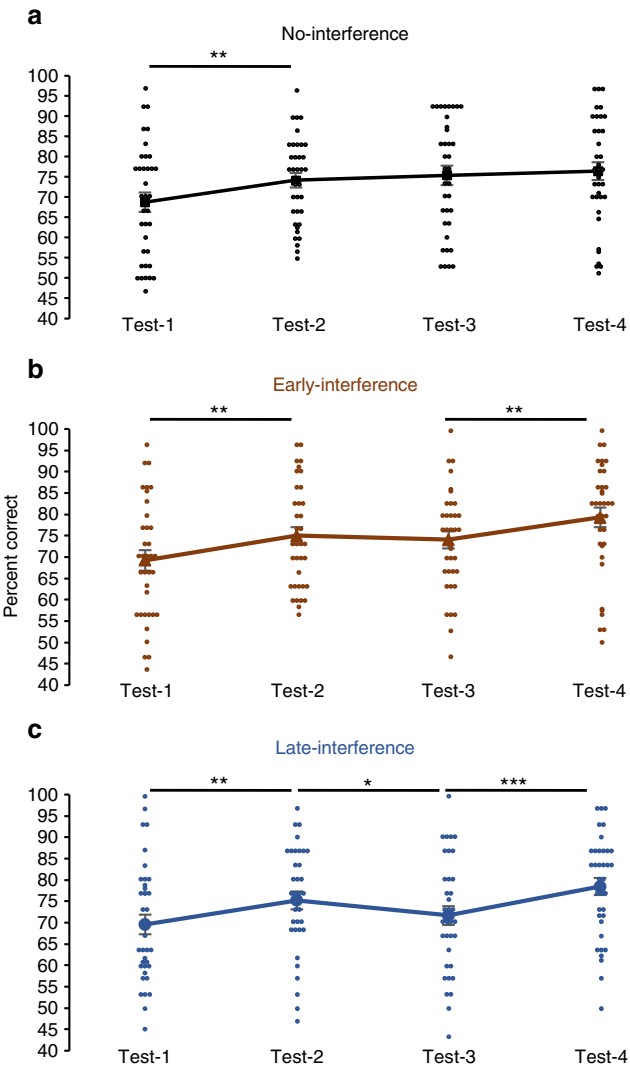

**Fig. 2** Auditory classification performance in experiment-1. The mean percentage of trials responded to correctly in the four posttests is shown for the **a** no-interference, **b** early-interference, and **c** late-interference conditions. Testing took place at 12:00 p.m. on day-1 (test-1), at 12:00 p.m. on day-2 (test-2), at 5:30 p.m. on day-2 (test-3), and at 12:00 p.m. on day-3 (test-4). The same 36 starlings completed each condition. Asterisks indicate significant differences between results at consecutive test times (*$P < 0.05$; **$P < 0.01$; ***$P < 0.001$ after Holm–Bonferroni corrections). Error bars show standard error of the mean. Dot plots show individual scores for each test session

memory should cycle through stable and labile states as the memory is retrieved and then reconsolidated. Does the pattern of use-dependent destabilization and sleep-dependent reconsolidation observed in experiment-1 continue across multiple bouts of retrieval and sleep?

Performance accuracy averaged across all conditions on the immediate post-training classification task-A test was 63.5 ± 1.7 (mean ± SEM), which was significantly greater than chance performance of 50% ($t_{23} = 7.82$, $P < 0.0001$, one-sample $t$-test, $n = 24$). Likewise, performance accuracy averaged across the six interference sessions from the two interference conditions for the post-training classification tasks B, C, and D tests was 64.7 ± 1.8, also significantly above chance ($t_{23} = 8.32$, $P < 0.0001$, one-sample $t$-test, $n = 24$). Immediate post-training test performance for each task did not differ across the conditions ($F_{2,46} = 1.13$, $P = 0.33$,

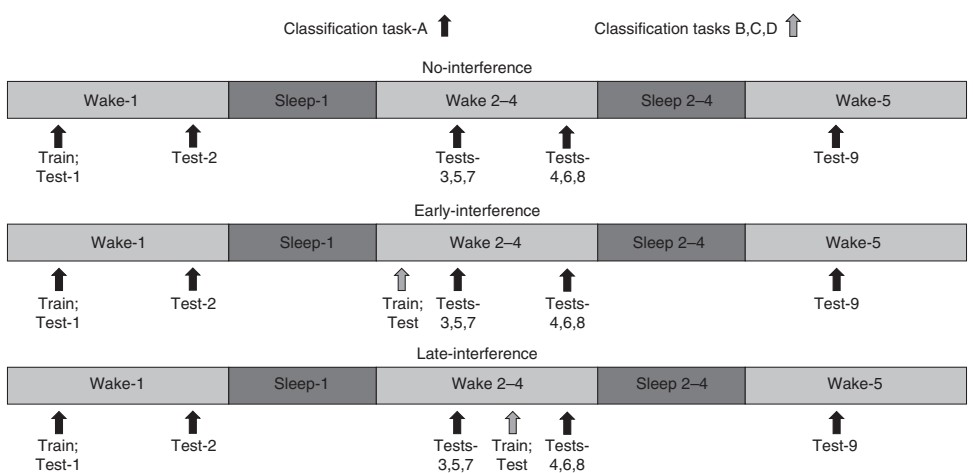

**Fig. 3** Experimental design for experiment-2. Twenty-four starlings were trained to classify 5 s segments of novel starling song. Starlings were trained on classification task-A from 10:00 a.m. to 12:00 p.m. and tested nine times thereafter: at 12:00 p.m. and 6:00 p.m. on days 1–4 and again at 12:00 p.m. on day-5. In the interference conditions, starlings also were trained and tested on classification task-B (on day-2), task-C (on day-3), and task-D (on day-4), either in the morning from 7:00 a.m. to 10:00 a.m. before task-A was tested (early-interference) or in the afternoon from 1:00 p.m. to 4:00 p.m. after task-A was tested (late-interference). The interference tasks (B, C, and D) entailed the same Go/No-Go operant training and testing procedure as task-A, but each task utilized different pairs of novel song stimuli. Each starling completed all three conditions

repeated-measures ANOVA, $n = 24$ for task-A and $t_{23} = 1.34$, $P = 0.19$, paired $t$-test, $n = 24$ for tasks B, C, and D). To evaluate performance on classification task-A following training and retention, we conducted a 3 (condition: no-interference, early-interference, late-interference) × 9 (test: tests 1–9) repeated-measures ANOVA ($n = 24$). While there were no significant effects for condition ($F_{2,46} = 1.02$, $P = 0.37$), there were main effects for test ($F_{8,184} = 18.00$, $P < 0.0001$) and the condition × test interaction ($F_{16,368} = 1.68$, $P = 0.05$).

Task-A classification performance in the no-interference condition (i.e., retention without task B, C, or D interference) decreased by a nonsignificant $1.9 \pm 2.3$ percentage points across the first day ($t_{23} = 0.85$, $P = 0.40$, paired $t$-test, $n = 24$) but increased significantly by $6.9 \pm 2.3$ percentage points after the first night ($t_{23} = 3.07$, $P = 0.006$, paired $t$-test, $n = 24$). Performance then showed gradual changes of $-1.4 \pm 1.8$, $2.3 \pm 2.2$, and $2.8 \pm 1.9$ percentage points over the three subsequent waking retention periods (days 2–4) and of $0.8 \pm 2.3$, $1.2 \pm 1.6$, and $2.7 \pm 1.4$ percentage points over the three subsequent sleeping retention periods. These represent nonsignificant changes of $1.2 \pm 1.4$ ($t_{23} = 0.88$, $P = 0.39$, paired $t$-test, $n = 24$) and of $1.6 \pm 1.1$ ($t_{23} = 1.41$, $P = 0.17$, paired $t$-test, $n = 24$) percentage points averaged across waking retention days 2–4 and sleeping retention nights 2–4, respectively (Figs. 4a and 5). By comparison, classification performance in the early-interference condition (i.e., interference tasks B, C, and D before task-A tests on days 2, 3, and 4) decreased by a non-significant $0.8 \pm 2.0$ percentage points across the first day ($t_{23} = 0.42$, $P = 0.68$, paired $t$-test, $n = 24$) but increased significantly by $6.4 \pm 2.1$ after the first night ($t_{23} = 2.98$, $P = 0.007$, paired $t$-test, $n = 24$). As in the no-interference condition, classification performance then showed changes of $-0.1 \pm 1.6$, $-2.4 \pm 2.3$, and $-2.2 \pm 2.7$ percentage points over the next three waking intervals for a nonsignificant average reduction of $1.6 \pm 1.3$ percentage points ($t_{23} = 1.24$, $P = 0.23$, paired $t$-test, $n = 24$). Unlike the no-interference condition, performance showed improvements of $5.5 \pm 1.7$, $5.0 \pm 2.5$, and $4.3 \pm 2.8$ percentage points over the next three sleeping intervals for a significant average improvement of $4.9 \pm 1.4$ percentage points ($t_{23} = 3.59$, $P = 0.002$, paired $t$-test, $n = 24$) (Figs. 4b and 5). Classification performance in the late-interference condition (i.e., interference tasks B, C, and D after task-A tests on days 2, 3, and

4) showed a non-significant change of $0.0 \pm 2.2$ percentage points across the first day ($t_{23} = 0.01$, $P = 0.99$, paired $t$-test, $n = 24$) but increased significantly by $6.9 \pm 1.7$ after the first night ($t_{23} = 4.20$, $P = 0.0003$, paired $t$-test, $n = 24$). In contrast to the no-interference and early-interference conditions, performance changed by $-9.6 \pm 2.6$, $-6.4 \pm 2.7$, and $-6.7 \pm 2.9$ percentage points over the subsequent waking periods for a significant average performance loss of $7.6 \pm 1.8$ percentage points ($t_{23} = 4.34$, $P = 0.0002$, paired $t$-test, $n = 24$). Nonetheless, performance improved over nights 2–4 by $11.7 \pm 2.6$, $7.6 \pm 2.2$, and $8.2 \pm 2.7$ percentage points, for a significant average improvement of $9.2 \pm 1.8$ percentage points across the subsequent sleeping retention periods ($t_{23} = 4.99$, $P < 0.0001$, paired $t$-test, $n = 24$) (Figs. 4c and 5).

Overall, there were no differences in performance changes across the conditions for the first wake retention period or the first sleep retention period. Each condition expressed non-significant changes across wakefulness ($F_{2,46} = 0.23$, $P = 0.80$, repeated-measures ANOVA, $n = 24$) and significant improvements across sleep ($F_{2,46} = 0.03$, $P = 0.97$, repeated-measures ANOVA, $n = 24$), confirming the pattern of stable performance after the first day of wakefulness without interference and enhanced performance after the first night of sleep established in our prior work[15,19]. However, the conditions expressed a different pattern of performance changes across waking retention on days 2–4 ($F_{2,46} = 7.82$, $P = 0.001$, repeated-measures ANOVA, $n = 24$) and across sleeping retention on nights 2–4 ($F_{2,46} = 0.23$, $P = 0.003$, repeated-measures ANOVA, $n = 24$). Across wakefulness on days 2–4, the late-interference condition showed significantly larger performance losses than the no-interference ($t_{23} = 3.26$, $P = 0.003$, paired $t$-test, $n = 24$) or the early-interference ($t_{23} = 2.70$, $P = 0.01$, paired $t$-test, $n = 24$) conditions, whereas the difference between the no-interference and early-interference conditions did not reach significance ($t_{23} = 1.70$, $P = 0.10$, paired $t$-test, $n = 24$). Across sleep on nights 2–4, the late-interference condition had significantly greater improvement than the no-interference condition ($t_{23} = 3.33$, $P = 0.003$, paired $t$-test, $n = 24$), and the early-interference condition showed marginally greater improvement than the no-interference condition ($t_{23} = 2.18$, $P = 0.04$, paired $t$-test, $n = 24$). The improvements in the early-interference and late-interference conditions were not

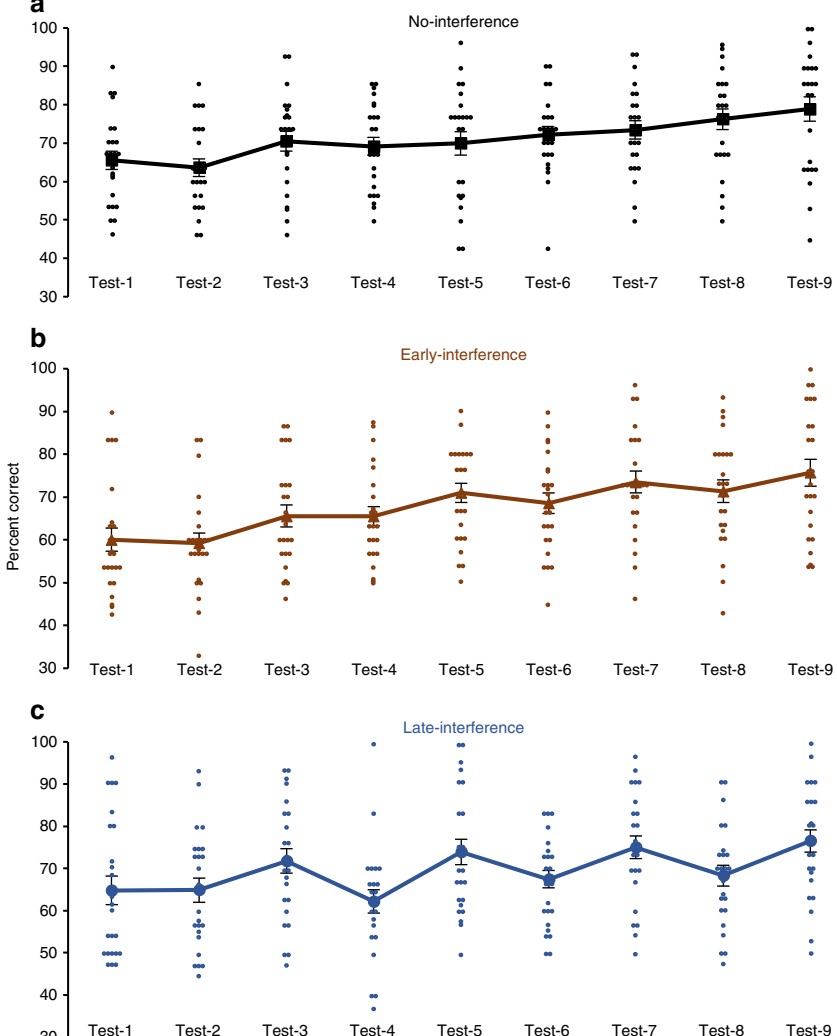

**Fig. 4** Auditory classification performance in experiment-2. The mean percentage of trials responded to correctly in the nine posttests is shown for the **a** no-interference, **b** early-interference, and **c** late-interference conditions. Odd-numbered tests took place at 12:00 p.m., and even-numbered tests took place at 6:00 p.m. The same 24 starlings completed each condition. Error bars show standard error of the mean. Dot plots show individual scores for each test session

significantly different ($t_{23} = 1.75$, $P = 0.09$, paired $t$-test, $n = 24$) (Fig. 5).

Finally, we conducted a 3 (condition: no-interference, early-interference, late-interference) × 2 (test: test-1, test-9) repeated-measures ANOVA ($n = 24$) to examine the overall performance change across the 5-day experiment. There was a significant main effect for test ($F_{1,23} = 38.71$, $P < 0.0001$), as the three conditions improved by $13.4 \pm 3.0$ (no-interference), $15.6 \pm 3.5$ (early-interference), and $11.7 \pm 3.8$ (late-interference) percentage points. However, there was not a significant condition ($F_{2,46} = 1.08$, $P = 0.35$) or condition × test interaction ($F_{2,46} = 0.37$, $P = 0.69$), indicating no difference in the overall performance gains across the 5 days. Nonetheless, the pattern of performance changes that led to comparable gains across the 5 days was noticeably different (Fig. 4a–c). Ultimately, starlings in the interference conditions were able to acquire multiple classification memories over the course of the experiment, regardless of time of day or time relative to task-A retrieval, and long-term memory formation was not impoverished compared to the condition that encountered no interfering experiences.

**New learning promotes sleep enhancement of related memories.** Starlings in each condition from experiments 1 and 2 expressed significant sleep-dependent performance improvements after the first night. Although performance increased after sleep, additional nights of sleep did not necessarily lead to additional performance gains. The effect of subsequent sleep on performance depended on the condition. Whereas the no-interference condition showed non-significant changes, classification performance in the two interference conditions both expressed significant gains across sleep. This suggests that the learning of the interference classification tasks after day-1 may have instantiated additional nights of sleep-dependent processing, resulting in memory improvements for task-A after interference learning and sleep. Do additional learning experiences, even those that can produce interference, promote the sleep-dependent processing of a similar learning experience? To directly test this question, experiment-3 was conducted as a simplified version of experiment-1 in which classification task-A was only tested on days 1 and 3, and classification task-B was trained on day-2 in the absence of any explicit task-A retrieval (Fig. 6).

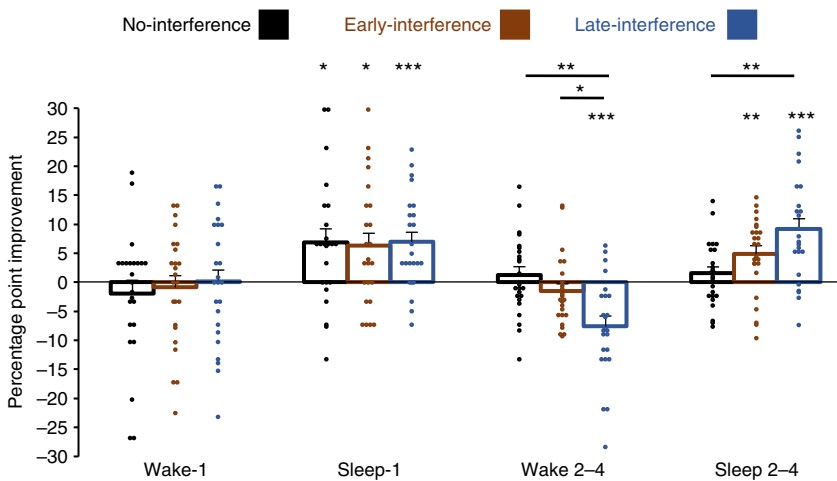

**Fig. 5** Performance improvement scores in experiment-2. The mean percentage-point improvement is shown for each condition grouped by the retention period: Wake-1 depicts the performance change across the first day of consolidation, Sleep-1 depicts the performance change across the first night of consolidation, Wake 2–4 depicts the average improvement scores over the three reconsolidation waking periods, and Sleep 2–4 depicts the average improvement scores over the three reconsolidation sleeping periods. The same 24 starlings completed each condition. Asterisks directly above the bars indicate significant performance changes for that specific condition and retention period. Asterisks above connecting lines indicate significant differences in performance changes between conditions within a given retention period (*$P < 0.05$; **$P < 0.01$; ***$P < 0.001$ after Holm–Bonferroni corrections). Error bars show standard error of the mean. Dot plots show individual percentage point improvement scores across the given retention period

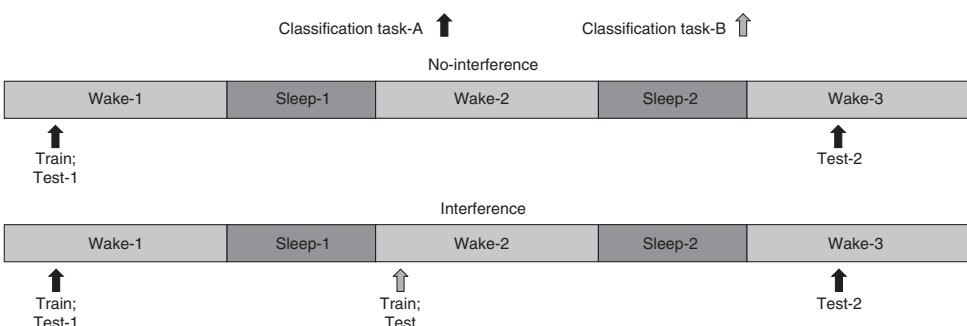

**Fig. 6** Experimental design for experiment-3. Fifty-seven starlings were trained to classify 5 s segments of novel starling song. Starlings were trained on classification task-A from 9:30 a.m. to 12:00 p.m. and tested two times thereafter: immediately after training at 12:00 p.m. on day-1 and 48 h later at 12:00 p.m. on day-3. In the interference condition, starlings also were trained and tested on classification task-B on day-2 from 9:30 a.m. to 1:00 p.m. Task-B entailed the same Go/No-Go operant training and testing procedure as task-A but utilized a different pair of novel song stimuli than task-A. Each starling completed both conditions

Performance accuracy averaged across both conditions on the post-training classification task-A test was $72.7 \pm 1.7$. This performance level was significantly greater than chance performance of 50% ($t_{56} = 13.29$, $P < 0.0001$, one-sample t-test, $n = 57$). Likewise, performance on the post-training classification task-B test for the interference condition was $75.1 \pm 2.1$, also significantly above chance ($t_{56} = 11.88$, $P < 0.0001$, one-sample t-test, $n = 57$). Immediate post-training test performance for the task-A test did not differ between the conditions ($t_{56} = 0.46$, $P = 0.65$, paired t-test, $n = 57$).

To evaluate task-A performance following training and retention after two nights of sleep, we conducted a 2 (condition: no-interference, interference) × 2 (test: tests 1 and 2) repeated-measures ANOVA ($n = 57$). While there was no significant effect for condition ($F_{1,56} = 0.59$, $P = 0.45$), significant effects for test ($F_{1,56} = 36.00$, $P < 0.0001$) and the condition × test interaction ($F_{1,55} = 7.1$, $P = 0.01$) were found. Task-A classification performance in the no-interference condition (i.e., retention without task-B training on day-2) showed a significant gain of $3.2 \pm 1.3$ from the immediate post-training test on day-1 to the post-

retention test on day-3 ($t_{56} = 2.47$, $P = 0.02$, paired t-test, $n = 57$). By comparison, task-A performance in the interference condition improved significantly by $8.0 \pm 1.3$ ($t_{56} = 6.25$, $P < 0.0001$, paired t-test, $n = 57$). Notably, the performance improvement in the interference condition was significantly greater than in the no-interference condition ($t_{56} = 2.66$, $P = 0.01$, paired t-test, $n = 57$) (Fig. 7).

We further investigated whether new auditory classification learning can benefit the memory of a recently learned classification task by testing task-A performance on days 1 and 5. For experiment-4, the no-interference condition only completed task-A, whereas starlings in the interference condition were trained on classification tasks B (day-2), C (day-3), and D (day-4) without explicit retrieval of task-A (Fig. 8). Performance accuracy averaged across both conditions on the post-training classification task-A test was $64.1 \pm 2.2$, which was significantly greater than chance ($t_{29} = 6.33$, $P < 0.0001$, one-sample t-test, $n = 30$). Likewise, average performance on the post-training classification task B, C, and D tests for the interference condition was $64.0 \pm 2.0$, also significantly above chance ($t_{29} = 7.31$, $P < 0.0001$, one-sample

*t*-test). Immediate post-training test performance for the task-A test did not differ between the conditions ($t_{29} = 0.97$, $P = 0.34$, paired *t*-test, $n = 30$).

To evaluate performance on classification task-A following training and retention after four nights of sleep, we conducted a 2 (condition: no-interference, interference) × 2 (test: tests 1 and 2) repeated-measures ANOVA ($n = 30$). While there were no significant effects for condition ($F_{1,29} = 0.31$, $P = 0.58$) or the condition × test interaction ($F_{1,29} = 1.61$, $P = 0.21$), there was a significant effect for test ($F_{1,29} = 18.00$, $P = 0.0002$). Task-A classification performance in the no-interference condition (i.e., retention without task B, C, or D training) showed a significant gain of $3.9 \pm 1.8$ from the immediate post-training test on day-1 to the post-retention test on day-5 ($t_{29} = 2.15$, $P = 0.04$, paired *t*-test, $n = 30$). By comparison, task-A performance in the interference condition improved significantly by $7.4 \pm 2.1$ ($t_{29} = 3.61$, $P = 0.001$, paired *t*-test, $n = 30$). The amount of improvement over 5 days was not different between the conditions ($t_{29} = 1.28$, $P = 0.21$, paired *t*-test, $n = 30$) (Fig. 9). Nonetheless, the magnitude of improvement was nearly double in the interference condition, despite having learned three additional classification tasks during the intervening period.

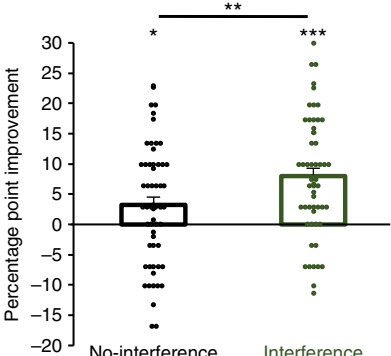

**Fig. 7** Auditory classification performance improvement in experiment-3. The mean percentage-point improvement from day-1 to day-3 is shown for both conditions. The same 57 starlings completed each condition. Asterisks directly above bars indicate significant performance changes across the 3-day retention interval within a condition. Asterisks above connecting lines indicate significant differences in the amount of improvement between conditions (*$P < 0.05$; **$P < 0.01$; ***$P < 0.001$ after Holm–Bonferroni corrections). Error bars show standard error of the mean. Dot plots show individual percentage point improvement scores from day-1 to day-3

## Discussion

We have demonstrated in starlings that auditory memories that have been consolidated by sleep can be returned to a labile state by retrieving the memories, thus making them vulnerable to subsequent interference. Sleep then reconsolidated the destabilized memories, and the observed cycle of use-dependent destabilization and sleep-dependent reconsolidation repeated across multiple days even as overall task performance improved. The results support a role for sleep in memory reconsolidation, and this conclusion is reinforced when considering alternative interpretations and methodological concerns. We considered if the putative sleep-dependent performance gains represent time-dependent changes because the waking intervals (12:00 p.m. to 5:30 or 6:00 p.m.) were shorter than the intervals with sleep (5:30 or 6:00 p.m. to 12:00 p.m. on the next day). Yet, our prior results in starlings[15] and an extensive literature on sleep and memory[8,10], including nap[21–23], and sleep deprivation[24–26] studies, have not supported a time-dependent explanation for memory benefits observed after sleep. Nevertheless, given the current results suggesting sleep-dependent reconsolidation, it will be important for future studies to confirm this effect in a modified design with similar waking and sleeping intervals. We also considered whether each testing session provided feedback that could act as a short training session. A role for sleep would be diminished if the sleep-dependent gains had already materialized during the prior testing session. Examining the learning dynamics within test sessions, however, did not provide consistent support that significant learning was evident within a test session (Supplementary Figs. 2–4). Finally, given that the largest sleep-dependent improvements tended to occur in the late-interference conditions, where performance had been reduced during the prior waking retention period, this suggests that the sleep benefit was largely compensating for prior performance losses. Yet, the correlations between daytime performance changes and the subsequent nighttime performance changes did not indicate a consistent pattern relating sleep-dependent gains to prior waking performance changes. Indeed, significant correlations were not limited to the late-interference condition, but rather significant and nonsignificant correlations were found in each condition (Supplementary Figs. 5–7).

Though the role of sleep in memory consolidation has been studied extensively[8–10], a role for sleep in memory reconsolidation has garnered little attention, and with ambiguous results. A prominent study of human motor-sequence learning[27] is commonly cited as evidence for sleep-dependent reconsolidation. In that study, participants (in groups 7 and 8) were trained on sequence-A on day-1, retested on sequence-A on day-2 (i.e., the

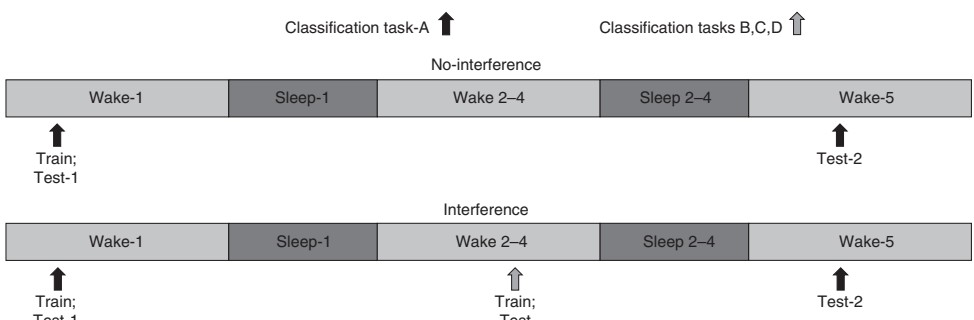

**Fig. 8** Experimental design for experiment-4. Thirty starlings were trained to classify 5 s segments of novel starling song. Starlings were trained on classification task-A from 10:00 a.m. to 12:00 p.m. and tested two times thereafter: immediately after training at 12:00 p.m. on day-1 and 96 h later at 12:00 p.m. on day-5. In the interference conditions, starlings also were trained and tested on classification task-B (on day-2), task-C (on day-3), and task-D (on day-4) from 1:00 p.m. to 4:00 p.m. The interference tasks (B, C, and D) entailed the same Go/No-Go operant training and testing procedure as task-A, but each task utilized different pairs of novel song stimuli. Each starling completed both conditions

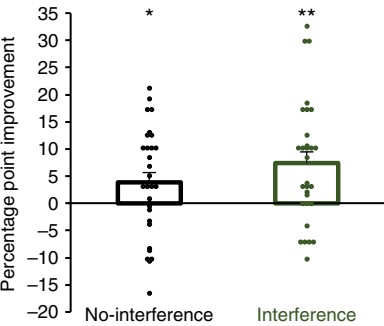

**Fig. 9** Auditory classification performance improvement in experiment-4. The mean percentage-point improvement from day-1 to day-5 is shown for both conditions. The same 30 starlings completed each condition. Asterisks indicate significant performance changes across the 5-day retention interval within a condition (*$P < 0.05$; **$P < 0.01$ after Holm–Bonferroni corrections). Error bars show standard error of the mean. Dot plots show individual percentage point improvement scores from day-1 to day-5

memory for sequence-A was reactivated), and then trained on sequence-B. Performance on sequence-A was stable when group-7 was retested immediately after learning sequence-B. In contrast, sequence-A accuracy was significantly impaired when group-8 was retested on day-3. While this shows that reactivating sequence-A can destabilize the sleep-consolidated memory of sequence-A in a time-dependent manner, it does not demonstrate that sleep reconsolidated sequence-A because task performance was significantly worse after sleep. Moreover, a role for sleep in the performance change from day-2 to day-3 cannot be determined because the 24-h retention interval included both wakefulness and sleep, and the performance level prior to sleep was not tested. Interpretation of this result becomes more problematic because the methods that were used in the study have been shown to produce fatigue-related confounds[18,28]. Furthermore, a recent study with multiple replication attempts failed to reproduce the reported destabilization effect[29].

Two other human studies have provided mixed evidence relating sleep to reconsolidation. In one study, a 40-min nap following reactivation of a visuo-spatial memory helped to stabilize the memory against subsequent interference. However, a similar effect was found even without memory reactivation, and there was no difference in post-nap performance between the reactivated ("reconsolidation group") and non-reactivated ("remote consolidation group") conditions[30]. In a separate study of syllable-pair learning, no behavioral difference in memory retention between the "labile" and "stable" conditions was found after sleep[31].

Rodent studies have also provided mixed evidence relating sleep to reconsolidation. Reactivation of a morphine place preference memory followed by 6 h of sleep deprivation led to impaired performance when tested the following day[32]. Antidepressant suppression of REM sleep also produced gradual impairments of a familiar memory over 5 consecutive days[33]. The implication of this work is that sleep is important for reconsolidation because depriving or suppressing sleep led to performance impairments. Yet, impairing memory through sleep deprivation or pharmacological sleep suppression is not equivalent to showing that natural sleep consolidates or reconsolidates a memory. Indeed, the relationship between sleep and reconsolidation is even more opaque because the transition to REM sleep state (i.e., TR-sleep) was found to correlate negatively with memory[33]. Thus, a state of unmanipulated sleep was associated with impaired reconsolidation.

A persistent confound for relating sleep to reconsolidation has been that reconsolidation studies have often used 24-h retention periods. Without testing task performance prior to sleep, one cannot differentiate memory changes that occur during the post-reactivation waking period before sleep from changes that occur during sleep. In contrast, the present study tested classification performance of the destabilized memory after both waking and sleeping retention over multiple days. Whereas performance was impaired across periods of waking retention if interference was encountered after task-A retrieval, the task-A memory then recovered and stabilized after sleep. Thus, sleep in starlings appears to play a similarly important role in reconsolidation as it does in the original consolidation of the memory.

The results from experiment-3 revealed an unexpected finding regarding the interaction between recently learned memories. Namely, learning classification task-B on the day after learning task-A led to greater sleep-dependent performance improvements for task-A than when task-B was not encountered. This is particularly striking because the inclusion of task-B was previously used to impair task-A performance[19,20]. Yet here it is acting to enhance the memory for task-A, apparently through a sleep-dependent mechanism because the learning of task-B did not improve task-A performance on the day of task-B learning but only after sleep. This unexpected result received further support in the more challenging design of experiment-4, which entailed a 5-day retention with multiple potentially interfering learning experiences.

How could learning task-B improve the memory for task-A after sleep? We have previously shown that learning task-B instantiates a night of sleep consolidation for task-B[19]. If the mechanisms of sleep consolidation and reconsolidation are similar, and the memories are related (e.g., due to similarity in stimuli, context, or task characteristics), then the mechanisms underlying the sleep-dependent consolidation of task-B could potentially benefit the memory for task-A as well. It is known that an episode of learning can alter the characteristics of subsequent sleep. For example, different types of learning have been shown to increase the amounts of slow wave sleep[34], REM sleep[35,36], and specific sleep features, such as thalamocortical spindles[37] and hippocampal sharp wave ripples[38]. Accordingly, the learning of task-B on day-2 could instantiate sleep-dependent mechanisms on the second night, which could then act to both consolidate task-B and reconsolidate task-A.

This could explain the results from experiments 1 and 2 in which the memory for task-A was presumably destabilized in each condition by each task-A posttest, yet significant performance gains after each night were only observed in the interference conditions. The performance improvements in the interference conditions in experiments 3 and 4 were also of a greater magnitude than the no-interference conditions even though experiments 3 and 4 were designed to not explicitly reactivate the task-A memory. However, the learning of the interference tasks may partially reactivate the memory of task-A. The stimuli utilized in both tasks consisted of starling songs. Although each song stimulus was novel, starling songs maintain a characteristic sequential organization, are composed of the same categories of sounds, and are recognized as conspecific songs[39]; consequently, learning to classify one song may reactivate memories of other songs recently learned under similar operant conditions.

It seems plausible that mechanisms induced by learning one classification task could affect a similar task if the tasks activate similar networks. The neural representation of auditory classification memories in starlings are known to be expressed in the caudal mesopallium[40–42]. Preliminary results have found that new classification learning increases the response strength of these auditory neurons to songs from a previously learned task (D. Zaraza and D. Margoliash, unpublished observations),

indicating that learning one task can reactivate memories from a similar task. Therefore, interference learning could reactivate, and therefore destabilize, the memory for task-A, thus enabling the mechanisms of sleep consolidation that were induced by the interference tasks to likewise benefit the destabilized memory of task-A. More broadly, this opens the possibility of coordinating training regimens with sleep to take advantage of sleep and memory interactions in order to optimize the consolidation of related memories.

The present work demonstrates that long-term memory formation in starlings, as also observed in several species of mammals including humans, entails a process of consolidation, destabilization, and reconsolidation, with sleep playing a critical role in both consolidation and reconsolidation. In complex natural conditions, such as when juvenile songbirds commit to memory a single song with special salience, reconsolidation may help to stabilize that memory even in the presence of interfering songs from various potential tutors experienced over many days[43]. Reconsolidation processes may further play a key role in refining the complexity of emerging vocalizations during the sensorimotor phase of learning, a sleep-dependent process that leads to a more accurate tutor song copy[44,45].

Assuming consolidation and reconsolidation processes are common in mammals and birds, this brings to question how similar sleep-dependent memory processes arose. The two predominant explanations of sleep-dependent consolidation—the synaptic homeostasis hypothesis[46] and active systems consolidation theory[47]—both attribute the beneficial effect of sleep on memory to processes occurring during slow wave sleep. Slow waves in mammals represent the slow oscillations of cortical membrane potentials, which are produced and synchronized via extensive corticocortical projections[48]. Birds have also evolved extensive palliopallial (corticocortical) connectivity. By one hypothesis[49], the high degree of pallial interconnectivity in birds may explain why birds express both slow wave sleep and clear evidence of sleep-dependent memory processing[15,19,44,45,50–52].

To what degree physiological processes such as slow wave sleep represent convergent evolution needs to be evaluated in the context of the emerging consensus of avian forebrain structure. An extensive research program including early studies of the avian auditory system[53,54] has identified key homologies between avian and mammalian circuits, recognizing that most of the avian forebrain is not of striatal origin[55,56]. Subsequent studies have identified molecular-specific cell types and circuits in the avian forebrain that have been described as homologous to those found in neocortex[57]. The resolution of these observations remains a subject of active discussion, but they support the hypothesis that a similar pattern of forebrain circuitry and cell types in birds and mammals stems from a common amniote ancestor[58,59]. Indeed, the turtle forebrain shares attributes of this pattern of forebrain connectivity but is distributed in fields with a three-layered cortex[57], and slow wave activity in a sleeping lizard was recently observed[60]. Overall, examining the similarities and differences of sleep and memory between mammals and birds represents a powerful approach for evaluating the functions of sleep, as well as understanding the relationship between complex sleep patterns, cortical connectivity, and memory processing[10,61,62].

## Methods

**Participants**. Ninety-one adult male and female European starlings (*Sturnus vulgaris*) were maintained on a 24 h schedule consisting of 14 h of light and 10 h of darkness (lights on at 6:00 a.m.). During the experiments, starlings were given free access to water at all times but were only given food via correct performance on an operant task (described below). Starlings were wild caught by USDA biologists at O'Hare airport in Chicago and transferred to the Margoliash Lab aviary. Prior to beginning the experiment, starlings were group housed in indoor flight aviaries with a light cycle approximating local time and with free access to food and water.

Once a starling was selected for an experiment, it was moved to an individual sound isolation box. The starling lived in a cage within the sound isolation box, and one wall of the cage consisted of the operant apparatus, which included the response ports and a food hopper. Thus, all experiments were conducted in the same location where the starling lived. All animal procedures were approved by the University of Chicago Institutional Animal Care and Use Committee (IACUC). Thirty-six of the starlings completed experiment-1. An additional 24 starlings completed experiment-1, but data from those 24 starlings were excluded from the analysis because a programming error produced incorrect reward contingencies for the task-B interference test and the task-A posttest-2 in the early-interference condition. The data from those 24 starlings for the no-interference and late-interference conditions, which did not include incorrect reward contingincies, is reported in Supplementary Fig. 1. Twenty-four different starlings completed experiment-2. Fifty-seven of the starlings from experiment-1 also completed experiment-3. Twenty-four of the starlings from experiment-2 as well as six additional starlings completed experiment-4. Sample sizes were chosen to be consistent with prior work[15,19,20].

**Stimuli**. Eighteen novel stimulus pairs, each consisting of two 5 s segments of natural starling song, were recorded from 18 starlings. One pair was used during two practice sessions before the experimental sessions began. Five pairs were used in the three conditions from experiment-1. Nine different pairs were used in the three conditions from experiment-2. Three different pairs were used in the two conditions from experiment-3. The same five pairs from experiment-1 were also used in the two conditions from experiment-4. A baseline stimulus pair that consisted of a rising tone (1–2 kHz) repeated three times versus a falling tone (3–2 kHz) repeated three times was used in the baseline classification task. The duration of each baseline stimulus was 1.7 s.

**Procedure**. Starlings completed an auditory classification task via a Go/No-Go operant procedure. Starlings initiated stimulus playback by probing a response port with their beak and had 2 s after stimulus completion to probe a second response port or withhold response. During each training session starlings could complete a maximum of 270 trials. Starlings initiated trials at their own pace, so starlings could complete a different number of trials in each session (see Supplementary Table 1 for data on number of trials completed per training and testing sessions). The training sessions for experiments 1 and 3 lasted 2.5 h. The training sessions for experiments 2 and 4 lasted 2 h. Responses to one stimulus (Go) produced a 4 s food reward; responses to the other stimulus (No-Go) resulted in a 15 s lights-out punishment. Nothing occurred if starlings withheld response. The stimulus for each trial was selected randomly, except that the same stimulus was selected for the next trial whenever a starling responded incorrectly, for up to three consecutive errors. After three consecutive errors, the stimulus for next trial was again selected randomly. Each test session lasted 1 h. Starlings received 15 min of free access to food at the beginning of each test session, and starlings could complete a maximum of 30 test trials during the remaining 45 min of the test session. Responses to the Go stimulus produced a 4 s food reward whereas responses to the No-Go stimulus resulted in a 15 s lights-out period. The stimulus for each trial was selected randomly (without the correction trials that were used during training) but constrained so that each stimulus was selected five times for each set of 10 trials. For each training and testing session, only one stimulus pair was presented per session, and each stimulus pair consisted of one Go stimulus and one No-Go stimulus.

Each starling became familiar with the Go/No-Go procedure by performing a baseline classification task with the baseline stimulus pair before starting the experimental conditions. Starlings were also engaged in the baseline task on experimental days whenever they were not completing training or testing sessions on the experimental stimuli, as well as on the days in between experimental conditions. Additionally, all starlings underwent two familiarization practice sessions by completing a training session with the practice stimulus pair from 7:45 to 9:45 a.m., followed by test sessions at 10:00 a.m., 2:30 p.m., and 6:00 p.m. for 2 days.

**Experimental design**. Each experiment entailed training on classification task-A, which involved discriminating between a pair of starling song stimuli using the Go/No-Go operant procedure. Task-A training was followed by an immediate post-training test on task-A as well as subsequent task-A tests. Experiment-1 included three additional task-A tests, experiment-2 included eight additional task-A tests, and experiments 3 and 4 included one additional task-A test. The interference conditions also entailed training on interference classification tasks (task-B for experiments 1 and 3 or tasks B, C, and D for experiments 2 and 4), as well as an immediate post-training interference test. The interference classification tasks utilized the same Go/No-Go procedure as task-A but involved starling song stimulus pairs that were different from those presented in task-A. The interference tasks were completed in the same cage with the same operant apparatus as task-A. For each experiment, the order of conditions was counterbalanced, and the stimulus pairs used in each condition were assigned randomly for each bird. The exact timing of the training and testing sessions for each experiment are presented in Supplementary Table 2.

In experiment-1 ($n = 36$), each starling completed three experimental conditions (Fig. 1), with three nights of sleep separating each condition. Starlings in each condition were trained on classification task-A on day-1 from 9:30 a.m. to 12:00 p.m. and received an immediate posttest at 12:00 p.m. This was followed by additional posttests on day-2 at 12:00 p.m. and 5:30 p.m., and on day-3 at 12:00 p.m. The no-interference condition only learned task-A. Two interference conditions included learning a second classification task-B on day-2. The early-interference condition included training and testing on task-B from 7:30 a.m. to 11:00 a.m. prior to the task-A posttest at 12:00 p.m. The late-interference condition included training and testing on task-B from 1:00 p.m. to 4:30 p.m. after the task-A posttest at 12:00 p.m. but before the task-A posttest at 5:30 p.m.

In experiment-2 ($n = 24$), each starling completed three experimental conditions (Fig. 3), with three nights of sleep separating each condition. Starlings in each condition were trained on classification task-A on day-1 from 10:00 a.m. to 12:00 p,m, followed by posttests at 12:00 p.m. and 6:00 p.m. on days 1–4 and a final posttest at 12:00 p.m. on day-5. The no-interference condition only learned task-A. Two interference conditions included learning three additional classification tasks on day 2 (task-B), day 3 (task-C), and day 4 (task-D). The early-interference condition included training and testing on the interference tasks from 7:00 a.m. to 10:00 a.m. prior to the task-A posttests at 12:00 p.m. on days 2–4. The late-interference condition included training and testing on the interference tasks from 1:00 p.m. to 4:00 p.m. after the task-A posttests at 12:00 p.m. but before the task-A posttest at 6:00 p.m. on days 2–4.

In experiment-3 ($n = 57$), each starling completed two experimental conditions (Fig. 6). Both conditions included training on classification task-A on day-1 from 9:30 a.m. to 12:00 p.m., followed by posttests at 12:00 p.m. on days 1 and 3. Whereas the no-interference condition only learned task-A, the interference condition included training and testing on a second classification task-B from 9:30 a.m. to 1:00 p.m. on day-2.

In experiment-4 ($n = 30$), each starling completed two experimental conditions (Fig. 8). Both conditions included training on classification task-A on day 1 from 10:00 a.m. to 12:00 p.m., followed by posttests at 12:00 p.m. on days 1 and 5. Whereas the no-interference condition only involved learning task-A, the interference condition included training and testing on three additional classification tasks from 1:00 p.m. to 4:00 p.m. on day-2 (task-B), day-3 (task-C), and day-4 (task-D).

**Performance measures**. Performance was measured as the percentage of correct trials during a test session, where a correct trial entailed responding to a Go stimulus or withholding response from a No-Go stimulus. Performance on the 50 baseline trials preceding each test session was also measured as a control for potential circadian confounds for experiments 1 and 2 because the test times in these experiments differed (Supplementary Figs. 8 and 9).

**Statistical analyses**. A two-way ANOVA with repeated measures on the test and condition factors was used to assess differences in classification accuracy and performance changes across the tests and conditions for each experiment. Performance changes across test sessions were analyzed using paired $t$-tests with the Holm–Bonferroni procedure to correct for multiple comparisons. The reported $P$ values are uncorrected, but statistically significant differences were determined using the appropriate $P$-value cutoff according to the Holm–Bonferroni procedure. One-sample $t$-tests were used to verify that post-training classification accuracy was greater than chance performance of 50%. All statistical tests were two-tailed, all data met test assumptions, and the variance was similar between groups being statistically compared. The experimenter was not blind to condition when assessing behavioral performance.

**Data availability**. The data that support the findings of these experiments are available from the corresponding author upon request.

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

## Acknowledgements

This work was supported by Grant N00014-12-1-0850 from Office of Naval Research to D.M. T.P.B. was supported by a Picower Fellowship from the Picower Institute for Learning and Memory during the writing of this manuscript.

## Author contributions

T.P.B. designed and conducted the experiments and analyzed the data. T.P.B., H.C.N., and D.M. wrote the manuscript.

## Additional information

**Competing interests:** The authors declare no competing interests.

