## [Peer Review File · Nature Communications]

Reviewers' comments:

Reviewer #1 (Remarks to the Author):

In their manuscript, Brawn et al. present exciting new finding regarding the interaction between sleep and reconsolidation. Using an operant discrimination task with starlings (task A), they show a sleep-related improvement in performance following an interference task (task B), but only when task B is preceded by a test that serves as a reminder of task A. Therefore, they conclude that destabilized memories benefit from sleep. Additionally, they show that the interference task, followed by sleep, improves performance in task A even when tasks A and B are decoupled.

The results in this paper are robust and intriguing and I believe they should be published in this journal. The authors do a good job in reproducing their own results in multiple experiments, adding to the manuscript's validity. However, some alternative explanations were not properly addressed and discussed.

Major points:

a. The manuscript repeatedly acknowledges the significant increase in memory following destabilization and interference. However, it is clear that these increases are highly correlated with the decreases they follow (i.e., in late-interference conditions, the 2nd test scores on the day(s) of interference are lower than the 1st test scores and most of the benefit from subsequent sleep is a compensation for this decrease). Obviously, the lower a score is, the more likely it is to go up after sleep - but there might also be other explanations for this finding. Perhaps there was a ceiling effect that prevented scores from constant inflation, except when those were lowered by destabilization. Perhaps the memory traces were not really altered during destabilization, but expression of this memory was blocked (and therefore the post-sleep results reflect mostly a return to baseline and not an improvement per se). This correlation between decreases in performance over the preceding day and benefit over the subsequent night deserves a more in-depth analysis. Reconsolidation and enhanced learning is only one interpretation of the data and sentences such as "whereas the no-interference condition showed non-significant changes, classification performance in the two interference conditions both improved significantly" (p.11) obscure this complexity, because in the net benefit over the full 24 hours was comparable between the two conditions.

b. A major limitation of this study is that it cannot fully distinguish between learning within an afternoon test session and learning over-night. Feedback is constantly provided throughout all tests. It could, therefore, be claimed, for example, that at least some of the improvement after destabilization and interference happened during the 5:30 PM task-A test (i.e., participants were very bad at the beginning of this test but used feedback to relearn and improved over the course of the session, before going to sleep). To reject this alternative explanation, the authors should show some of the dynamics within test sessions and, perhaps, show that the performance at the end and start of each session is comparable. If they're not, this should be addressed in the text.

c. On a number of occasions, the numbers reported in the main text do not match the ones displayed in the figures (e.g., page 6 refers to a data point with a value of 5.8, whereas the value in figure 2B is different; page 12 refers to a point with a value of 7.7, whereas the value in figure 7A is different). There may be more such mistakes which should be corrected in either the text or the figures.

Minor points:

a. In the abstract, the sentence starting with "We then provide" contains two unrelated findings that are based on different experiments. Presenting them together is misleading and confusing (i.e., the studies in which the destabilization manipulation was used are not the ones in which the interference

tasks promoted subsequent learning by themselves).

b. In the interpretation of the results of experiments 3 & 4, the authors consider the enhancement of the memory for task A as done "through a sleep-dependent mechanism" (Page 5). In fact, you can't be sure that it is not the interference task(s) itself, regardless of sleep, which was the cause of this effect. This ambiguity should be addressed directly in the manuscript.

c. The sentence on page 6 claiming that "sleep consolidated memory for task-A against task-B interference" is unclear (what does it mean to consolidate a memory of one task against the other?).

d. Two points regarding p-values in the paper:

1) In a number of occasions, the p-values in the main text appear as $p=X.XXX$ and not $P<X.XXX$, despite the fact that these comparisons are significant (i.e., should be $P<X.XXX$; e.g., page 8, $p=0.05$).

2) The correction for multiple comparisons is somewhat confusing. The authors claim that they judged significance based on corrected cutoffs, but some effects with p-value that are barely under 0.05 are still deemed significant. Additionally, the figures use stars to show significance, but, if I understand correctly, also incorporate the multi-comparison correction in some way. For example, in figure 2B the difference is shown as $p<0.01$ (and not $p<0.001$), even though in the text the p values appear as $p=0.0005$ and $p<0.0003$). The same goes for the two left black bars in figure 4 - and probably other instances too.

e. The large difference between the overall gain in performance for the different conditions in experiments 1 & 2 (as show in figure 6A) relative to their counterparts in experiments 3 & 4 (figure 7) requires some discussion. Do you think this is because of the different number of tests - and therefore feedback based training - between 1/2 and 3/4?

f. The authors discuss at length the possible mechanism by which engagement in task -B improved performance in task-A, including context effects. Were both task conducted in the same physical setting, which is different than where the starlings were housed?

g. The authors used repeated measures ANOVA in all but experiment 1, because of some data that has been omitted. However, a regular 2-way ANOVA cannot be used in experiment 1, because it assumes that data is independent, which it clearly is not (i.e., the same starlings participated in the different conditions). The correct way to conduct this analysis, I believe, would be to drop all the data (for all conditions, not just the early interference) for the 24 starlings whose data was omitted - and run a repeated measures ANOVA on all the remaining 36 starlings, as has been done for the other experiments. At the very least, if the decrease in N reduces the power of the experiment significantly, the authors should note that this analysis yields results in the same direction as the ones they reported.

h. There seems to be some mix-up in the Stimuli paragraph. I believe nine stimuli pairs were used for experiment 2 and three for experiment 3 and not the other way around.

i. For clarity, I suggest adding a sentence to the procedure explaining that in each session only one sound pair was used and that one of this pair's stimuli was a go sound and the other a no-go sound.

j. The reasoning for the subsequent presentation of a non-remembered stimulus (Procedure, p.20) is unclear. Will it keep being presented until the starling is correct once?

k. The procedure states that the starlings were engaged in the baseline task whenever they were not completing the training/testing. That is not very clear - did they hear those sounds in their home cage? How does this not contradict the food deprivation regime? The baseline data after each test is later said to have been used to measure circadian confounds. Why is this analysis not reported?

l. The naming of Task A/B/C etc. is somewhat confusing, given that the task is exactly the same, only the sound pair changes. It may be a good idea to consider changing this terminology to Stimuli Pair A or something similar. This would help clarify both the potential for interference and the rationale behind your suggestions about the mechanism by which Task-B contributed to performance in Task-A.

m. At the bottom of page 5, before "Post-training test performance" the authors should add "Immediate" for clarity (because all tests were done post-training).

Points concerning the figures:

- a. The figure layout in the manuscript is confusing. Figures 1A-D are interspersed with the other figures in a manner that made the order hard to understand. Perhaps each of the four experiments would benefit from its own figure, showing both the design (on one panel) and the results.
- b. The "Test"/"Train/Test" labels in the design figures sometimes make the text unintelligible (e.g., Fig 1A, bottom, center). It may be helpful to substitute these with different arrows types or colors.
- c. It may be useful to use the same names for the tests in the text and figures. The text refers to "posttest-N", and it would be helpful if the design figures show which test is which. Specifically, this would provide a useful way to overcome confusing labels such as "Night 2" (Figure 2, 3, 6B), which counterintuitively does not refer to the sleep period, but to the test preceding it.
- d. All three y-axes in figure 3 should have the same limits.
- e. In figure 5, "ms" should probably be "ns" (?)

Reviewer #2 (Remarks to the Author):

This is an impressive and well-controlled study that examined the time course of repeated memory destabilization and reconsolidation in starlings over several days, including periods of sleep and wakefulness. The findings show markedly different patterns of memory destabilization upon reactivation and subsequent reconsolidation depending on the time point of interference learning as well as the retention period spanning a day of wakefulness or a night including sleep.

The study design is sophisticated and elegant and the methods seem to be sound. The findings are novel and highly interesting and clearly advance our understanding of the time course and determinants of consolidation and reconsolidation processes. I am rather enthusiastic about the data and would like to congratulate the authors on a very nice paper. However, I have a few concerns mainly with regard to the interpretation of the data and some methodological details.

1. Although the data nicely demonstrate different time courses of destabilization and reconsolidation in the different conditions, I am not convinced that the observed effects are dependent on sleep. As is now, the sleep and wake intervals are not comparable at all. They differ fundamentally in the length of the retention interval (18 hours across sleep, only 6 hours across wake) and the temporal proximity to the reactivation/interference events. In the late interference condition, the wake period even includes interference learning. Thus, the observed differences could be explained either by i) intervening sleep or ii) the simple passage of time or iii) proximity to reactivation/interference or iv) any combination of these factors. In order to show that the observed effects are specific for sleep, there would have to be wake control intervals with exactly the same timings as in the sleep intervals, i.e. with an 18-hour retention period, starting a few hours after interference learning. Importantly, I am not saying that the findings are not interesting in their present form. Yet, they do not prove conclusively that the effects are dependent on sleep.

2. It is difficult to conclude that the interval across the night (i.e. what the authors call "Sleep") specifically benefited reconsolidation (in the late interference condition). As can be seen from Figure 2, there are similar improvements if the interference task is introduced before reactivation/destabilization. The improvements across "Sleep" in the late interference condition could reflect a boost by learning of task B similar to the early interference condition as well as to Experiments 3 and 4. In fact, this similarity seems to be reminiscent of the study by Klinzing et al (2016) that the authors discuss on p. 14, arguing that in this study "a similar effect was found even without memory reactivation". The same seems to be true in the present study. How can this be explained? Is the improvement from night 2 to day 3 comparable in the early and late interference conditions in Figure 2?

3. The results of Exp. 3 and 4 seem to correspond nicely to the results of Exp. 2. The posttest sessions on day 3 of Exp. 2 in the no-interference and early-interference conditions seem to correspond to Exp. 3, and the posttest sessions on day 5 of Exp. 2 in the no-interference and early-interference conditions seem to correspond to Exp. 4. Like in Exp. 3 and 4, also in Exp. 2 the difference between conditions on day 3 seems to be larger than on day 5, which may be due to a slower and more gradual increase of performance in the no-interference condition compared to the interference conditions. Thus, I am not clear why the authors conclude that Exp. 4 revealed "remarkably similar" findings as Exp. 3, given that the data from Exp. 2 in fact support the observation that the benefit from interference learning that is observed at day 3 becomes smaller over time and basically vanishes on day 5. Please report respective comparisons for Exp. 2 on days 3 and 5. This observation should also be discussed more adequately.
4. The timing of interference learning should be described in more detail. In the methods section, it only says that learning of the interference tasks took place "prior to the task-A posttest at 12:00 P.M." (in the early interference condition) and "after the task-A posttest at 12:00 P.M. but before the task-A posttest at 5:30 [or 6:00 in Exp. 2] P.M." (in the late interference condition). Considering that specific time windows have been proposed for destabilization and reconsolidation, it would be important to know the exact timing.
5. In the methods section, it says that starlings could complete "up to 270 trials" during each training session. What does "up to" mean here? Did single starlings differ in the number of trials? If so, please provide the range and average trials and report whether conditions were comparable in the mean number of training trials.
6. In the methods section, it is mentioned that the first 50 baseline trials after each test session were used to control for potential circadian confounds. These data should be reported to exclude circadian factors.
7. Did the authors test any of the interference tasks again? This would be interesting to see how performance on these tasks develops over time.
8. Figure 5 is redundant as it shows exactly the same data as Figure 4, only arranged differently. This figure should be removed. Likewise, Figure 6 is not really necessary as these data are basically evident from Figure 3.
9. The authors argue prominently that this study reports reconsolidation during sleep "for the first time". This is not correct, considering that other studies have tested this question before. Moreover, doing something for the first time is not per se a sign of quality. This phrase may be removed.

Reviewer #3 (Remarks to the Author):

Overall assessment:

This MS presents elegant experiments, demonstrating complex sleep dependent reconsolidation across destabilized memories. The most interesting aspect of the MS is the interactions across memories. Namely, learning a new task may temporarily impair performance of a previously learned task, but sleep might then improve the performance of both. This finding raises questions (which should be discussed) about the dynamics of complex learning, perhaps also in a sensory-motor context.

Specific comments:

1. As I was reading the MS I kept wondering about possible 'side effects' of testing sessions on memories. Of course, testing sessions are designed to minimize such effects, but no details were presented in the Methods section about the testing procedure to allow judging it. Measurements always change the measured features, this is unavoidable. However, here this issue is of particular concern since conclusions depend on the assumption that testing does not induce further learning. The

concern is that 'second night' is also a 'first night' after learning induced by the testing. Providing some more details about the testing and of possible 'side effects' of those sessions could help.

2. Figure numbers are not properly ordered: 1A then 2, then 1B... Please make figure numbers linear.

3. Figure 2b: classification performance at posttest-4 showed a significant improvement of 5.3 ± 1.3 percentage points after a second night of sleep. But it is not clear if this second improvement resulted in a higher level of performance compared to birds who experienced no interference? It is difficult to judge if the second improvement in Figure 2b could mirror a recovery from the slight deterioration during previous day, or (more interestingly) a real additional improvement step triggered by interference. This can be tested statistically across the two groups. Perhaps try testing if the final performance differ across the two groups?

4. I would suggest adding a discussion section focusing on possible implications for complex sensory-motor learning, such as song learning. Sleep appears to affect such learning as well, inducing oscillations that are structurally similar to those reported here. Perhaps authors could outline some preliminary framework for an extended framework based on multiple, hierarchical learning tasks, could link between the current paper and natural sensory-motor learning?

Regarding comments of Reviewer 1:

In their manuscript, Brawn et al. present exciting new finding regarding the interaction between sleep and reconsolidation. Using an operant discrimination task with starlings (task A), they show a sleep-related improvement in performance following an interference task (task B), but only when task B is preceded by a test that serves as a reminder of task A. Therefore, they conclude that destabilized memories benefit from sleep. Additionally, they show that the interference task, followed by sleep, improves performance in task A even when tasks A and B are decoupled.

The results in this paper are robust and intriguing and I believe they should be published in this journal. The authors do a good job in reproducing their own results in multiple experiments, adding to the manuscript's validity. However, some alternative explanations were not properly addressed and discussed.

Major Points:

- 1. The manuscript repeatedly acknowledges the significant increase in memory following destabilization and interference. However, it is clear that these increases are highly correlated with the decreases they follow (i.e., in late-interference conditions, the 2nd test scores on the day(s) of interference are lower than the 1st test scores and most of the benefit from subsequent sleep is a compensation for this decrease). Obviously, the lower a score is, the more likely it is to go up after sleep - but there might also be other explanations for this finding. Perhaps there was a ceiling effect that prevented scores from constant inflation, except when those were lowered by destabilization. Perhaps the memory traces were not really altered during destabilization, but expression of this memory was blocked (and therefore the post-sleep results reflect mostly a return to baseline and not an improvement per se). This correlation between decreases in performance over the preceding day and benefit over the subsequent night deserves a more in-depth analysis. Reconsolidation and enhanced learning is only one interpretation of the data and sentences such as "whereas the no- interference condition showed non-significant changes, classification performance in the two interference conditions both improved significantly" (p.11) obscure this complexity, because in the net benefit over the full 24 hours was comparable between the two conditions.**

The reviewer comments that the significant increase in memory after sleep that follows destabilization and interference in the late-interference condition is highly correlated with the memory decreases that occurred *during* the preceding waking period. To evaluate this relationship, we ran a linear regression for each condition correlating the performance changes over the daytime retention period with the performance changes over the subsequent sleep period. This included examining the performance changes across day-2 and night-2 for experiment-1 as well as the performance changes across day-2, day-3, and day-4 with night-2, night-3, and night-4, respectively, for experiment-2. As shown in the tables below, the correlations are significant in some cases but not in others. Notably, there is not a consistent pattern indicating this effect is specific to the late-interference condition since each condition shows both significant and non-significant correlations for different day/night retention pairs.

Scatterplots of these data are now presented in Supplementary Figs. 5-7, and this peer review response is referenced in the manuscript text.

No-Interference	r²	P
Day-2/Night-2 (exp-1; n=36)	0.26	0.002
Day-2/Night-2 (exp-2; n=24)	0.11	0.11
Day-3/Night-3 (exp-2; n=24)	0.09	0.14
Day-4/Night-4 (exp-2; n=24)	0.002	0.85

Early-Interference	r²	P
Day-2/Night-2 (exp-1; n=36)	0.02	0.47
Day-2/Night-2 (exp-2; n=24)	0.0002	0.94
Day-3/Night-3 (exp-2; n=24)	0.55	<0.0001
Day-4/Night-4 (exp-2; n=24)	0.35	0.003

Late-Interference	r²	P
Day-2/Night-2 (exp-1; n=36)	0.14	0.02
Day-2/Night-2 (exp-2; n=24)	0.14	0.07
Day-3/Night-3 (exp-2; n=24)	0.001	0.86
Day-4/Night-4 (exp-2; n=24)	0.41	0.007

The reviewer notes that a lower evening score will be more likely to go up after sleep. This seems plausible and may explain some of the correlations that do exist across the conditions. However, a performance score that was reduced in the evening due to interference, as in the late-interference condition, would not necessarily be expected to increase after sleep. One of the most prominent studies on sleep and reconsolidation (Walker et al., *Nature*, 2003) observed that the learning of an interfering motor sequence after reactivating the target sequence led to impaired performance accuracy on the following day (after sleep). This indicates that the interference may have disrupted reconsolidation and task performance did not benefit from sleep. Indeed, a highlight of the current work is that sleep reconsolidates memory despite interference-induced impairments.

The reviewer raised a concern that there is a ceiling effect “that prevented scores from constant inflation except when those were lowered by destabilization”. We do not believe this is the case. First, we have found that starlings are capable of performing at very high levels on these auditory classification tasks. The performance levels in the current experiments (~65-80% correct) do not represent an upper bound on their abilities, so it does not seem that a ceiling effect is preventing them from further increases. Second, the early-interference condition, on average, showed significant performance gains after sleep that were not accompanied by significant prior daytime losses, suggesting that sleep-dependent gains are not dependent on prior losses. Third, as the correlations above show, there was not a consistent pattern across day/night retention periods relating daytime performance losses to sleep-dependent gains.

The reviewer suggests the possibility that the memory traces were not altered when they were destabilized by reactivation but rather that the expression of the memory was blocked. Given that these experiments did not include electrophysiology or imaging, we cannot be certain what happened to the memory traces during the reactivation, destabilization, and

reconsolidation periods. However, destabilization (i.e., returning the memory to a labile state) of the memory traces would have occurred in each condition due to task-A retrieval. If the memory traces were inaccessible because they were blocked, task performance would be expected to fall to chance. Yet, performance at the evening retests in the no-interference and early-Interference conditions did not show impaired performance, indicating that the memories were not blocked. Task performance in the late-interference condition was also still significantly above chance even after the interference-induced performance impairments. This suggests that destabilization alone does not block expression of the memories. If we take “blocked” to be a graded rather than all-or-none phenomenon, then the impaired performance after reactivation and interference in the late-interference condition could obtain from reduced access to the memory at the evening retest. Whether the interference reduces access to the original memory or alters it remains an open question.

Here we provide a framework for how interference may affect recently acquired, related memory traces. During task-A learning, memory engrams underlying the associations between each task-A stimulus and its appropriate response would form. These engrams would be reactivated during the subsequent task-A tests. The engrams would become labile again from the reactivation but the stimulus-response mappings would remain unchanged in the absence of an interfering task, leading to stable behavioral performance. During task-B learning, additional associations would be formed between the task-B stimuli and responses. Some neurons in the task-B engrams might overlap with neurons in the task-A engrams because memories encoded closely in time tend to have overlapping representations (e.g., Rogerson et al., *Nature Reviews Neuroscience*, 2014; Cai et al., *Nature*, 2016). In this case, the temporal proximity would connect the memory reactivation of task-A with the memory encoding of task-B. Neurons can also be recruited to engrams based on their level of neuronal excitability immediately before training (Yiu et al., *Neuron*, 2014). Thus, neurons that were excited during the task-A retest may be preferentially recruited as members of the task-B representations. Consequently, overlapping representations could produce retroactive interference because some neurons that had signaled “go” in task-A could become members of engrams that signal “no-go” in task-B. Hence, the same neurons could be members of ensembles that signal opposite responses, thus weakening the original stimulus-response mapping and leading to reduced performance on a subsequent task-A retest after waking retention. During sleep, mechanisms of consolidation/reconsolidation would act to strengthen the memories. These mechanisms could help to recover what was lost. For example, as suggested by the synaptic homeostasis hypothesis, synaptic downscaling induced by slow waves could help to remove the weaker synapses connecting the task-A and task-B engrams that may have caused the interference, leading to stimulus-response mappings with a stronger signal-to-noise ratio after sleep. Additionally, the memories could be strengthened above their previous levels via a reactivation mechanism that specifically potentiates the memory traces during sleep replay. Clearly, this framework is speculative and future experiments will be needed to address if and how the memory traces change during the post-activation waking period and the subsequent sleeping period.

- 2. A major limitation of this study is that it cannot fully distinguish between learning within an afternoon test session and learning over-night. Feedback is constantly provided throughout all tests. It could, therefore, be claimed, for example, that at least some of the improvement after destabilization and interference happened during the**

5:30 PM task-A test (i.e., participants were very bad at the beginning of this test but used feedback to relearn and improved over the course of the session, before going to sleep). To reject this alternative explanation, the authors should show some of the dynamics within test sessions and, perhaps, show that the performance at the end and start of each session is comparable. If they're not, this should be addressed in the text.

We agree with the reviewer that the feedback provided during testing could allow for learning during the test sessions, and therefore it is important to evaluate the learning dynamics within test sessions. We have examined these learning dynamics by splitting all the test sessions into two halves and using paired *t*-tests to compare performance during the first and second halves. The results are now reported in Supplementary Figs. 2-4, and we have referenced this peer review response in the manuscript text. The results indicate that there is not a consistent pattern that test performance improved during testing. Of the 47 test sessions across the 4 experiments, performance on the second half of the test was higher in 27 sessions and lower in 20 sessions. Importantly, only 4 of the 47 tests showed a significant difference ($P < 0.05$) between the first and second halves, and 3 more showed a marginally significant difference ($P < 0.10$). Note that these statistics do not include corrections for multiple comparisons. The results also do not support a pattern of within test sessions learning when looking specifically at test performance in the late-interference conditions, indicating that the performance gains across the sleep period had not materialized prior to sleep. Overall, the analysis of within test session learning does not refute the experimental results indicating a role for sleep in reconsolidation.

- 3. On a number of occasions, the numbers reported in the main text do not match the ones displayed in the figures (e.g., page 6 refers to a data point with a value of 5.8, whereas the value in figure 2B is different; page 12 refers to a point with a value of 7.7, whereas the value in figure 7A is different). There may be more such mistakes which should be corrected in either the text or the figures.**

We thank the reviewer for catching these errors and apologize for the oversight. We have corrected these discrepancies and verified that the numbers reported are now correct and consistent across the text and figures.

Minor Points:

- 4. In the abstract, the sentence starting with "We then provide" contains two unrelated findings that are based on different experiments. Presenting them together is misleading and confusing (i.e., the studies in which the destabilization manipulation was used are not the ones in which the interference tasks promoted subsequent learning by themselves).**

We have revised the abstract to make clear that these are separate findings from different experiments.

- 5. In the interpretation of the results of experiments 3 & 4, the authors consider the enhancement of the memory for task A as done "through a sleep-dependent mechanism" (Page 5). In fact, you can't be sure that it is not the interference task(s)**

itself, regardless of sleep, which was the cause of this effect. This ambiguity should be addressed directly in the manuscript.

The reviewer suggests that the increased improvement in the interference conditions for experiments 3 and 4 could be due to the interference learning itself regardless of sleep. In contrast, our interpretation is that the increased improvement in the interference conditions resulted from the combination of the interference learning and sleep. The experimental results do not support the claim that the interference learning enhanced the task-A memory without sleep. If learning task-B was able to improve the memory for task-A without sleep, then one would expect task-A performance to be higher during the day following task-B training. However, the early-interference condition in experiment-1 learned task-B and was then tested on task-A at two points later in the day. If the learning of task-B by itself improved the memory for task-A, then task-A performance at test-2 (12:00 P.M.) should be higher than the no-interference or late-interference conditions (which had not learned task-B at that point), but the performance improvements on task-A from day-1 to day-2 were comparable in all conditions. Furthermore, task-A performance, which was tested several hours later, had still not improved by test-3 (5:30 P.M.) in the early-interference condition. Yet, performance was enhanced after a night of sleep. Alternatively, task-B training in the late-interference condition led to reduced rather than improved performance when tested on the same day as task-B, but task-A performance was enhanced after sleep. Critically, a second night of sleep did not improve task-A memory if task-B was not learned (as in the no-interference condition). Accordingly, after the first night of consolidation, memory improvements for task-A only occurred if both an interference task was learned and starlings had a night of sleep. This strongly suggests that the increased improvement in the interference conditions of experiments 3 and 4 occurred due to interference learning and sleep, and not by the interference learning itself. We have added a sentence to the discussion to clarify this point, and we have referenced this peer review response in the manuscript text.

- 6. The sentence on page 6 claiming that "sleep consolidated memory for task-A against task-B interference" is unclear (what does it mean to consolidate a memory of one task against the other?).**

This sentence has been revised in the manuscript. Our intended meaning was that the memory for task-A had been stabilized by sleep. Since it was in a stable state after sleep, it was not susceptible to disruption from the task-B interference that was encountered after the first night of sleep.

- 7. In a number of occasions, the p-values in the main text appear as $p=X.XXX$ and not $P<X.XXX$, despite the fact that these comparisons are significant (i.e., should be $P<X.XXX$; e.g., page 8, $p=0.05$).**

The P values are reported with equalities because the Nature Communications instructions state "provide exact values for both significant and non-significant P values". For P values less than 0.0001, we reported as $P < 0.0001$ because the software we used did not report exact P values lower than that.

8. **The correction for multiple comparisons is somewhat confusing. The authors claim that they judged significance based on corrected cutoffs, but some effects with p-value that are barely under 0.05 are still deemed significant. Additionally, the figures use stars to show significance, but, if I understand correctly, also incorporate the multi-comparison correction in some way. For example, in figure 2B the difference is shown as $p < 0.01$ (and not $p < 0.001$), even though in the text the p values appear as $p = 0.0005$ and $p < 0.0003$). The same goes for the two left black bars in figure 4 - and probably other instances too.**

We use the Holm-Bonferroni method for multiple comparison corrections within a family of tests. In a Bonferroni correction, the alpha level is divided by the total number of comparisons within a family (α/n). If alpha is 0.05 and there are 3 comparisons, the P value for each test must be less than $0.05/3 = 0.167$ to be considered significant at the $\alpha = 0.05$ level. In a Holm-Bonferroni correction, the test with the smallest (i.e., most significant) P value must be less than α/n . If that test reaches significance, the test with the next smallest P value must be less than $\alpha/(n-1)$, and so on. Once a non-significant test is reached, all remaining tests are non-significant. For example, in the late-interference condition in experiment-1, 3 tests were conducted examining the change from test-1 to test-2, test-2 to test-3, and test-3 to test-4. The largest change was from test-3 to test-4 ($t_{35} = 4.35$, $P = 0.0001$), which is significant at $\alpha = 0.001$ because 0.0001 is less than $0.001/3 = .0003$. The 2nd largest change was from test-1 to test-2 ($t_{35} = 3.69$, $P = 0.0008$), which is significant at $\alpha = 0.01$ because 0.0008 is less than $0.01/2 = .005$ but it is not significant at the $\alpha = 0.001$ because 0.0008 is greater than $0.001/2 = 0.0005$. Finally, the last comparison was from test-2 to test-3 ($t_{35} = 2.35$, $P = 0.02$), which is significant at the $\alpha = 0.05$ because 0.02 is less than $0.05/1 = 0.05$. Overall, this method controls family-wise error rate but is more powerful than a simple Bonferroni correction. We recognize that there may be confusion because Nature Communications asks for exact P values for all tests, and therefore the P values reported in text are often lower (i.e., more significant) than the corresponding asterisks used in the figures.

9. **The large difference between the overall gain in performance for the different conditions in experiments 1 & 2 (as show in figure 6A) relative to their counterparts in experiments 3 & 4 (figure 7) requires some discussion. Do you think this is because of the different number of tests - and therefore feedback based training - between 1/2 and 3/4?**

The reviewer comments that the overall performance change is larger in experiments 1 and 2 compared to experiments 3 and 4. This observation is mostly, but not entirely, correct. This difference in performance gains is most noticeable when comparing the 5-day experiments from experiments 2 and 4. For example, the no-interference condition in experiment-2 improved by 13.4 ± 3.0 from day-1 to day-5, whereas the no-interference condition in experiment-4 improved by 3.9 ± 1.8 over the same period. As the reviewer suggests, this difference is likely due to the repeated testing that occurred in experiment-2 but not in experiment-4. The intervening tests between day-1 and day-5 in experiment-2 provided those starlings up to 210 additional trials (up to 30 trials per test over 7 additional tests) as well as frequent reminders that can also be beneficial to long-term memory

formation. The results showed incremental gains, regardless of waking or sleeping retention, that averaged 1.4 ± 0.5 with each test. That represents a gain of 9.8 percentage points, which, if added to the experiment-4 improvement of 3.9, would result in a nearly identical improvement of 13.7 compared to the 13.4 of experiment-2.

For the interference conditions, the early-interference and late-interference conditions from experiment-2 showed improvements of 15.6 ± 3.5 and of 11.7 ± 3.8 , respectively, whereas the improvement in the interference condition from experiment-4 was 7.4 ± 2.1 over the same period. The additional testing in experiment-2 likely played a role in this difference as the testing provided additional feedback and reminders. But the testing also served to destabilize the memory, which allowed the interference to reduce performance across waking retention that then had to be overcome during sleep. The overall larger gain for the interference conditions in experiment-2 therefore likely resulted from the combination of repeated testing, smaller (early-interference) or larger (late-interference) performance decreases across waking, and smaller (early-interference) or larger (late-interference) performance increases after sleep. In contrast, performance on task-A in the interference condition in experiment-4 appeared to receive a sleep benefit from the first interference task (task-B on day-2) but perhaps not as much from the second and third interference tasks (task-C on day-3 and task-D on day-4). Indeed, the performance gain of 7.4 ± 2.1 from day-1 to day-5 was similar to the performance gain of 8.0 ± 1.3 from day-1 to day-3 in the interference condition from experiment-3 that only received task-B interference training on day-2.

When comparing the overall performance gains in experiments 1 and 3, a similar result is obtained for the no-interference conditions. The extra test sessions (and perhaps a slightly greater sleep-dependent gain) led to an improvement of 7.7 ± 2.0 in experiment-1 compared to an improvement of 3.2 ± 1.3 in experiment-3. In contrast, the overall improvement for the interference conditions in experiment-1 (10.1 ± 1.5 for early-interference and 8.9 ± 2.1 for late-interference) was similar to the overall improvement for the interference condition in experiment-3 (8.0 ± 1.3). This similarity seems to result from the fact that there were only two additional tests in experiment-1 compared to experiment-3 and that the interference task-B appeared to produce comparable sleep-dependent benefits for task-A in both experiments.

Please also see our response to point-23 below that covers a related point comparing performance gains across the different experiments. We have referenced this peer-review response in the manuscript text.

10. The authors discuss at length the possible mechanism by which engagement in task-B improved performance in task-A, including context effects. Were both tasks conducted in the same physical setting, which is different than where the starlings were housed?

The starlings lived in the experimental apparatus, so the task location and housing location were the same. Prior to experiments, starlings lived in our lab aviary with other starlings. Once a starling was chosen for an experiment, it was moved to an individual sound isolation box that contained a cage. One wall of the cage consisted of the operant apparatus, which included the response ports for initiating and responding to trials and the food hopper that became available to the starlings when they responded correctly to Go stimuli. Once a starling was moved into a sound box, it was not moved until it had completed all of the experimental conditions. The training and testing for task-A was identical to the training and

testing of the interference tasks (B, C, and D). Thus, starlings completed all the experiments in the same context in which they lived. This information has been added to the methods.

- 11. The authors used repeated measures ANOVA in all but experiment 1, because of some data that has been omitted. However, a regular 2-way ANOVA cannot be used in experiment 1, because it assumes that data is independent, which it clearly is not (i.e., the same starlings participated in the different conditions). The correct way to conduct this analysis, I believe, would be to drop all the data (for all conditions, not just the early interference) for the 24 starlings whose data was omitted – and run a repeated measures ANOVA on all the remaining 36 starlings, as has been done for the other experiments. At the very least, if the decrease in N reduces the power of the experiment significantly, the authors should note that this analysis yields results in the same direction as the ones they reported.**

We have revised the manuscript according to the reviewer's suggestion to only report results from the 36 starlings that have full data sets in experiment-1 and to analyze the data with a repeated-measures 2-way ANOVA as we did for experiments 2-4. We have now reported the results from the 24 starlings in experiment-1 that had incomplete data sets in Supplementary Fig. 1. These results serve as a replication of the pattern of results reported in the main text for experiment-1.

- 12. There seems to be some mix-up in the Stimuli paragraph. I believe nine stimuli pairs were used for experiment 2 and three for experiment 3 and not the other way around.**

The reviewer is correct. Nine stimulus pairs were used in experiment-2 and three stimulus pairs were used in experiment-3. This has been revised.

- 13. For clarity, I suggest adding a sentence to the procedure explaining that in each session only one sound pair was used and that one of this pair's stimuli was a go sound and the other a no-go sound.**

We have clarified this in the procedure section of the methods.

- 14. The reasoning for the subsequent presentation of a non-remembered stimulus (Procedure, p.20) is unclear. Will it keep being presented until the starling is correct once?**

During each training session, the stimulus for a given trial was selected randomly with the exception that we also used "correction trials", which meant that the same stimulus was chosen for the next trial if the starling responded incorrectly to the current trial. This same stimulus was chosen for up to 3 consecutive errors. As an example, the stimulus for Trial-1 in a given training session was chosen randomly. If the starling responded incorrectly to Trial-1, the stimulus for Trial-2 was the same as Trial-1. If the starling responded incorrectly to Trial-2, the stimulus for Trial-3 was the same as Trial-1 (and Trial-2). If the starling responded incorrectly again, the stimulus for Trial-4 would be chosen randomly because the limit of 3 consecutive errors had been reached. In sum, the stimulus for the next trial was randomly

chosen if the starling responded correctly to the previous trial or if the starling had responded incorrectly to the previous 3 trials. We used correction trials during training because it helps to correct response biases and helps the starlings learn the stimulus-response associations more rapidly. The correction trials, however, were not used during the test sessions because the test sessions were designed to present an equal number of Go and No-Go stimuli. We have clarified this in the procedure section of the methods.

15. The procedure states that the starlings were engaged in the baseline task whenever they were not completing the training/testing. That is not very clear - did they hear those sounds in their home cage? How does this not contradict the food deprivation regime?

The starlings lived in the cage where the experiments were conducted, so they were not moved between a home cage and an experiment cage. There was also not a specific food deprivation regime. Rather, starlings received all their food by responding correctly to the Go stimuli while running the Go/No-Go task. This was true for both the experimental days and the days in between the conditions (as well as during the pre-training before the experiment). The baseline Go/No-Go task was initiated each day as soon as the cage-lights went on at 6:00 A.M. The starlings were extensively trained on the baseline task, so getting food via the baseline task was easily accomplished. The only aspect of the experiment that could resemble a food restriction regime would be during the training and testing sessions. Since food was available only by correct response to Go stimuli, starlings would have less access to food during training and testing because the number of trials was limited and because they had to learn the stimulus-response associations. Since the starlings may have received less food than they were otherwise used to during the training sessions, we preceded each test session with 15 min of access to free food. This was intended to ensure that performance on the immediate post-training test was not confounded by the starlings being overly hungry. Overall, starlings gained access to food via the baseline task from 6:00 A.M. (lights-on) to 8:00 P.M. (lights-off) during the days before the experiment began and on the days in between each condition once the experiment had started. Starlings also ran the baseline task on experiment days during the hours when they were not being trained or tested on the experimental stimuli. The exact timing of when the experimental and baseline tasks took place for each condition is now reported in Supplementary Table 2.

16. The baseline data after each test is later said to have been used to measure circadian confounds. Why is this analysis not reported?

Experiments 1 and 2 involved comparing test sessions that took place at different times of the day (12:00 P.M. and 5:30 P.M. for experiment-1 and 12:00 P.M. and 6:00 P.M. for experiment-2). Because the testing occurred at different times of day, it is possible that performance changes were actually due to circadian effects on performance (e.g., classification performance could be inherently better or worse in the morning or in the evening). However, circadian effects seem unlikely as an explanation for the results because task-A test performance did not vary based on time of testing but rather on the inclusion and specific timing of an interference task. Moreover, task-A performance was similar at test-1 (12:00 P.M.) and test-2 (6:00 P.M.) on day-1 for each condition in experiment-2. Since

performance remained stable across these time points (in which interference was not encountered), it seems that performance is not affected by the timing of the tests.

Moreover, the starlings were engaged in the baseline classification task whenever they were not completing the experimental task. If circadian factors affected performance, then one would expect to see consistent changes in baseline performance depending on time of day. To test for this, we used the 50 baseline trials that preceded the test sessions as a marker of the baseline performance level. Test-1 for each condition was excluded from this analysis because test-1 was preceded by a 2.0- or 2.5-hour training session rather than the baseline task. For experiment-1, we ran a 3 (Condition) x 3 (Test: Tests 2-4) repeated-measures ANOVA on the baseline performance. For experiment 2, we ran a 3 (Condition) x 8 (Test: Tests 2-8) repeated-measures ANOVA on the baseline performance. There were no significant differences in baseline performance in either case, suggesting that task performance at the testing times was not affected by circadian factors. The baseline data are now reported in Supplementary Figs. 8-9.

- 17. The naming of Task A/B/C etc. is somewhat confusing, given that the task is exactly the same, only the sound pair changes. It may be a good idea to consider changing this terminology to Stimuli Pair A or something similar. This would help clarify both the potential for interference and the rationale behind your suggestions about the mechanism by which Task-B contributed to performance in Task-A.**

We agree that there is some potential for confusion regarding the task names. However, we have decided to keep the names as is. In considering other options, we believe it created awkward sentence constructions in some cases and did not necessarily alleviate potential for confusion. We would also like to keep the naming consistent with our published work on interference that uses this naming scheme. To help reduce any confusion, we have made more explicit in the text, figure legends, and methods that tasks A, B, C, and D all refer to same Go/No-Go task but with different stimulus sets.

- 18. At the bottom of page 5, before "Post-training test performance" the authors should add "Immediate" for clarity (because all tests were done post-training)**

“Immediate” has been added to the text.

- 19. The figure layout in the manuscript is confusing. Figures 1A-D are interspersed with the other figures in a manner that made the order hard to understand. Perhaps each of the four experiments would benefit from its own figure, showing both the design (on one panel) and the results. b. The "Test"/"Train/Test" labels in the design figures sometimes make the text unintelligible (e.g., Fig 1A, bottom, center). It may be helpful to substitute these with different arrows types or colors. c. It may be useful to use the same names for the tests in the text and figures. The text refers to "posttest-N", and it would be helpful if the design figures show which test is which. Specifically, this would provide a useful way to overcome confusing labels such as "Night 2" (Figure 2, 3, 6B), which counterintuitively does not refer to the sleep period, but to the test preceding it. d. All three y-axes in figure 3 should have the same limits. e. In figure 5, "ms" should probably be "ns" (?)**

We thank the reviewer for noticing some errors and suggesting improvements to the figures. We have revised the numbering of the figures. We have adjusted the sizing and spacing of the figure elements and text to increase clarity, including the “Test” and “Train/Test” labels. We have added test numbers to the figures where applicable, and the scale for the y-axes has been adjusted to be equal across the conditions within an experiment.

Regarding comments of Reviewer 2:

This is an impressive and well-controlled study that examined the time course of repeated memory destabilization and reconsolidation in starlings over several days, including periods of sleep and wakefulness. The findings show markedly different patterns of memory destabilization upon reactivation and subsequent reconsolidation depending on the time point of interference learning as well as the retention period spanning a day of wakefulness or a night including sleep.

The study design is sophisticated and elegant and the methods seem to be sound. The findings are novel and highly interesting and clearly advance our understanding of the time course and determinants of consolidation and reconsolidation processes. I am rather enthusiastic about the data and would like to congratulate the authors on a very nice paper. However, I have a few concerns mainly with regard to the interpretation of the data and some methodological details.

20. Although the data nicely demonstrate different time courses of destabilization and reconsolidation in the different conditions, I am not convinced that the observed effects are dependent on sleep. As is now, the sleep and wake intervals are not comparable at all. They differ fundamentally in the length of the retention interval (18 hours across sleep, only 6 hours across wake) and the temporal proximity to the reactivation/interference events. In the late interference condition, the wake period even includes interference learning. Thus, the observed differences could be explained either by i) intervening sleep or ii) the simple passage of time or iii) proximity to reactivation/interference or iv) any combination of these factors. In order to show that the observed effects are specific for sleep, there would have to be wake control intervals with exactly the same timings as in the sleep intervals, i.e. with an 18-hour retention period, starting a few hours after interference learning. Importantly, I am not saying that the findings are not interesting in their present form. Yet, they do not prove conclusively that the effects are dependent on sleep

The reviewer is correct that the different duration for the waking and sleeping retention intervals raises doubts as to whether the observed effects are due to sleep or other factors such as time. We would have preferred to use equal 12-hour retention intervals, as is typically used in human studies on sleep and memory, but there were pragmatic considerations that motivated the design of our animal study to use unequal retention intervals. The starling training/testing sessions in our experiments are much longer (3 or 3.5 hours) than the training/testing sessions often used in human studies (less than 1 hour). Consequently, using a 12-hour retention interval would have placed the evening tests into the

starling sleep period. Furthermore, starling behavior can become strongly driven by appetitive factors very early or very late in the day, especially when food is used as the operant reward as in our experiments. We thus avoid training and testing during those times. Additionally, the experiments required maintaining a consistent time for the task-A tests across multiple days, and the first task-A test of the day (12:00 P.M.) needed to be late enough to allow for the full interference training and testing session before the test (for the early-interference condition) and be early enough to allow for the full interference training and testing session after the first task-A test but before the evening task-A test (for the late-interference condition). These considerations led to the testing times that were used in these experiments. Although these limitations of our animal study are not shared with human studies, it was also possible to keep the animals under constant conditions and on a baseline task whenever not being trained or tested. This would be very difficult to achieve in human studies over multiple days but was highly instrumental in allowing us to arrive at strong conclusions regarding interference, sleep, and reconsolidation. We hope this too is taken into consideration. We fully agree that future work should attempt to confirm the sleep reconsolidation effect with a modified design that more closely equates the sleeping and waking retention intervals.

Despite these limitations, we believe sleep is the most plausible explanation for the gains that were observed over nighttime retention periods. First, our prior studies (Brawn et al., 2010a, 2013) with starlings utilized longer (9 hour) waking retention intervals. The results across waking are very similar to what was observed in the current experiments. Namely, task-A performance showed non-significant changes if there was no interference but showed performance losses if task-B was encountered after task-A. This indicates that a longer waking interval did not improve or recover the task-A memory. Yet, performance after the sleep retention interval resulted in significant improvements. Additionally, task-A performance suffered from task-B interference when task-B was learned immediately after task-A or hours later. This suggests that the negative impact of task-B interference did not depend on the relative timing of task-A and task-B. Second, a vast literature on the role of sleep in memory also does not support the time-dependent hypothesis. A multitude of human studies using 12-hour retention intervals have demonstrated that 12 hours of sleep consolidates a variety of memory tasks whereas 12 hours of wakefulness does not. Furthermore, nap studies have demonstrated that memory improvements can arise over shorter (i.e., less than 12 hours) retention periods if they include a nap and that daytime retention periods with naps produce memory benefits whereas the same retention periods without naps (i.e., waking retention) do not. Thus, nap studies have compellingly shown that it is not time per se, but the inclusion of sleep, that is necessary for a memory consolidation effect. Finally, the time-dependent explanation has also been rejected by sleep deprivation studies. Under the time-dependent hypothesis, memory should improve even during sleep deprivation because the passage of time, not sleep, would be responsible for consolidation. However, studies have shown that memory does not improve after sleep deprivation (even when attempting to control for deprivation-related confounds). Ultimately, the time-dependent explanation seems improbable given our prior published results and the extensive results on the role of sleep in memory processing. We have addressed this issue in the discussion and referenced this peer-review response in the manuscript text.

21. It is difficult to conclude that the interval across the night (i.e. what the authors call “Sleep”) specifically benefited reconsolidation (in the late interference condition). As can be seen from Figure 2, there are similar improvements if the interference task is introduced before reactivation/destabilization. The improvements across “Sleep” in the late interference condition could reflect a boost by learning of task B similar to the early interference condition as well as to Experiments 3 and 4. In fact, this similarity seems to be reminiscent of the study by Klinzing et al (2016) that the authors discuss on p. 14, arguing that in this study “a similar effect was found even without memory reactivation”. The same seems to be true in the present study. How can this be explained? Is the improvement from night 2 to day 3 comparable in the early and late interference conditions in Figure 2?

We do not think that sleep specifically benefitted reconsolidation only in the late-interference conditions from experiments 1 and 2. Rather, memory destabilization and subsequent reconsolidation would have occurred in each condition because the retrieval of task-A at the test sessions would have destabilized the task-A memory regardless of condition. Whether memory destabilization resulted in memory impairments depended on the occurrence and timing of the interfering experiences. Performance in the no-interference and early-interference conditions remained stable after the 12:00 P.M. test for the remainder of the day because 1) there were no subsequent interfering experiences to impair the newly destabilized memory and 2) the interference in the early-interference condition was encountered prior to the task-A test when the task-A memory would have been stable. In contrast, the interference in the late-interference condition was encountered after the memory destabilization, leading to impaired performance at the evening retest. The evening (5:30 or 6:00 P.M.) task-A test in each condition would further serve to destabilize the task-A memory. Therefore, going into the sleep period, the task-A memory was destabilized, and thus in need of reconsolidation, for each condition.

We propose that reconsolidation then occurred for each condition during sleep. The late-interference condition showed this the most clearly because sleep recovered what was lost due to the task-A memory reactivation and subsequent task-B interference. The early-interference condition also showed reconsolidation, which appeared behaviorally as improved performance after sleep. Experiment-2 further showed that the memory had become stable again because it was not impaired by the interference tasks on the third and fourth mornings. We believe that the task-A memory was also reconsolidated (i.e., re-stabilized) in the no-interference condition. Behaviorally, it appears as if little happened because performance only showed very slight increments over the day and night, but the other conditions strongly indicate that memory destabilization occurred from the task-A retest and that the reconsolidation only occurred after a sleep period. Nonetheless, verifying reconsolidation in the no-interference condition would require a modified design that challenged the task-A memory with an interference task on the third morning to show that it had become resistant to the interference.

The improvements from night-2 to day-3 in experiment-1 were comparable in the early- and late-interference conditions ($t_{35} = 0.69$; $P = 0.49$). Given that the task-A memory in each condition was in an unstable state going into sleep, why did the interference conditions show significant improvements across sleep whereas the no-interference condition did not? We propose that the learning of the interference task initiated sleep-dependent mechanisms that

enhanced the sleep-dependent improvements for task-A. Task learning can alter the characteristics of subsequent sleep, which can include increases to slow wave sleep, spindles, ripples, and replay. Indeed, we have a work in prep showing that classification learning in starlings leads to increased slow wave sleep early in the night that correlates with post-sleep improvements. We speculate that the learning of the interference tasks initiated sleep-dependent consolidation/reconsolidation mechanisms (e.g., increased slow waves that could lead to synaptic downscaling or increased replay) that served to both consolidate the interference task and reconsolidate the destabilized task-A memory. There would be no significant performance gains in the no-interference condition because there was no training during the day that would lead to enhanced sleep-dependent processes. Overall, we do not believe that reconsolidation was specific to the late-interference condition. Rather, the late-interference condition was crucial to the experiment not because it was the only condition to experience destabilization and reconsolidation, but because 1) it clarified that the task-A retest did in fact destabilize the memory (since interference was then able to impair the task-A memory), and 2) it showed that the destabilized and impaired memory could recover after sleep.

Finally, it seems plausible that the early-interference condition may represent a middle ground between the no-interference and late-interference condition because the training of task-B might partially reactivate the task-A memories since the Go/No-Go task was the same, the physical environment/context of the experiment was the same, and each stimulus consisted of species-specific vocalizations. This may explain why performance in the early-interference condition tended to get slightly worse over the day whereas performance in the no-interference condition tended to get slightly better. It may also be responsible for the results in experiments 3 and 4 where interference task-B training without task-A retrieval led to better task-A performance after sleep. This peer-review response has been referenced in the manuscript text.

22. The results of Exp. 3 and 4 seem to correspond nicely to the results of Exp. 2. The posttest sessions on day 3 of Exp. 2 in the no-interference and early-interference conditions seem to correspond to Exp. 3, and the posttest sessions on day 5 of Exp. 2 in the no-interference and early-interference conditions seem to correspond to Exp. 4. Like in Exp. 3 and 4, also in Exp. 2 the difference between conditions on day 3 seems to be larger than on day 5, which may be due to a slower and more gradual increase of performance in the no-interference condition compared to the interference conditions. Thus, I am not clear why the authors conclude that Exp. 4 revealed “remarkably similar” findings as Exp. 3, given that the data from Exp. 2 in fact support the observation that the benefit from interference learning that is observed at day 3 becomes smaller over time and basically vanishes on day 5. Please report respective comparisons for Exp. 2 on days 3 and 5. This observation should also be discussed more adequately.

Our claim that the results from experiments 3 and 4 were similar was based on the fact that both experiments showed a pattern of performance gains that was larger in the interference conditions than in the no-interference conditions. But we agree with the reviewer that the difference between the conditions in experiment-2 gets smaller over days 3-5, which results from the gradual increase in the no-interference condition. Indeed, the overall

performance change from day-1 to day-3 was 4.4 ± 2.4 for the no-interference condition but 10.9 ± 2.4 for the early-interference condition ($t_{23} = 2.27$; $P = 0.03$). Yet, by day-5, the overall performance changes were not different for the no-interference (13.4 ± 3.0) and early-interference (15.6 ± 3.5) conditions ($t_{23} = 0.54$; $P = 0.60$). The results from experiments 3 and 4 then also support the observation that the benefit from interference learning becomes smaller over time. We cannot know why the benefit from interference learning diminishes without knowing the mechanism, but it is plausible that the interference tasks on days 3 and 4 do not benefit task-A to the same extent that the interference task on day-2 does because they are further removed from task-A encoding and may not reactivate the task-A engrams as strongly. For example, it is possible that learning task-B on day-2 weakly reactivates the task-A memory and leads to sleep-dependent processes that consolidate the newly encoded task-B memory and reconsolidate the reactivated task-A memory. Learning task-C on day-3 may then weakly reactivate the task-B memory (and perhaps reactivate the task-A memory even less or not at all), leading to sleep consolidation of task-C and sleep reconsolidation of task-B over the third night. Learning task-D on day-4 may then weakly reactivate the task-C memory (and perhaps not reactivate the task A and B memories at all), leading to sleep consolidation of task-D and sleep reconsolidation of task-C over the fourth night. Further research will be needed to characterize the behavioral effects and the mechanisms of how interference and sleep can lead to these memory benefits.

Please also see our response to point-9 above that covers a related point comparing performance gains across the different experiments. We have deleted the phrase “remarkably similar” and referenced this peer-review response in the manuscript text.

- 23. The timing of interference learning should be described in more detail. In the methods section, it only says that learning of the interference tasks took place “prior to the task-A posttest at 12:00 P.M.” (in the early interference condition) and “after the task-A posttest at 12:00 P.M. but before the task-A posttest at 5:30 [or 6:00 in Exp. 2] P.M.” (in the late interference condition). Considering that specific time windows have been proposed for destabilization and reconsolidation, it would be important to know the exact timing.**

We have now reported in Supplementary Table 2 a series of tables displaying the exact timing of the tasks completed in each experiment. We have also added more timing information to the methods and figure legends.

- 24. In the methods section, it says that starlings could complete “up to 270 trials” during each training session. What does “up to” mean here? Did single starlings differ in the number of trials? If so, please provide the range and average trials and report whether conditions were comparable in the mean number of training trials.**

Starlings initiated trials at their own pace during the training and testing sessions. The maximum number of trials allowed was 270 during the training sessions and 30 during the test sessions. We have now reported in Supplementary Table 1 the mean and standard deviation of the number of trials run in each training and testing session. Within each experiment, the number of training trials was comparable across conditions. There were more training trials completed in experiments 1 and 3 compared to experiment 2 and 4, but this

makes sense given that the training sessions for experiments 1 and 3 were 30 minutes longer than for experiments 2 and 4 (2.5 hours vs. 2.0 hours). There also tended to be more training trials completed in the late-interference tasks compared to the early-interference tasks in experiments 1 and 2. This likely resulted from the fact that starlings run fewer trials early in the morning, and the early-interference conditions started at 7:30 A.M. (Experiment-1) or 7:00 A.M. (Experiment-2). Nonetheless, the number of trials completed in the interference tasks for the early-interference conditions was not significantly different than the number of trials completed in the corresponding late-interference conditions ($t_{35} = 1.8$, $P = 0.08$ for experiment-1 and $t_{23} = 1.3$, $P = 0.21$ for experiment-2). Performance on the interference tests was also not significantly different between the early-interference and late-interference conditions ($t_{35} = 0.82$, $P = 0.42$ for experiment-1 and $t_{23} = 1.71$, $P = 0.10$ for experiment-2). Finally, the number of test trials completed were comparable in all conditions.

25. In the methods section, it is mentioned that the first 50 baseline trials after each test session were used to control for potential circadian confounds. These data should be reported to exclude circadian factors.

Please see our response to point-16 above.

26. Did the authors test any of the interference tasks again? This would be interesting to see how performance on these tasks develops over time.

The interference tasks were not retested at any point in these experiments. However, we did retest the interference task in our previous publication “Sleep consolidation of interfering auditory memories in starlings” (Brawn et al., *Psychological Science*, 2013) as well as in “Differential development of retroactive and proactive interference during post-learning wakefulness” (Brawn et al., *Learning & Memory*, 2018), which is currently in press. In the first study, in which both task-A and task-B were trained on day-1, experiment-1 examined the memory for task-A and experiment-2 examined the memory for task-B. The results from that study showed that both tasks interfered with each other across waking retention, resulting in impaired performance on both tasks when they were retested in the evening. That is, the learning of task-B retroactively interfered with the memory for task-A, and the learning of task-A proactively interfered with memory for task-B. However, sleep consolidated the memory for both tasks despite the interference. Performance on task-A and task-B were significantly improved when they were retested the next day after a night of sleep. In the second study, task-A and task-B were trained on the same day and retested at different points later in the day. That study showed an asymmetry in the emergence of the two types of interference such that task-A performance was impaired immediately after learning task-B, whereas the task-B impairment required additional time to emerge.

27. Figure 5 is redundant as it shows exactly the same data as Figure 4, only arranged differently. This figure should be removed. Likewise, Figure 6 is not really necessary as these data are basically evident from Figure 3.

We have combined figures 4 and 5 into a single figure (now figure 5) and have removed what was figure 6 from the manuscript.

- 28. The authors argue prominently that this study reports reconsolidation during sleep “for the first time”. This is not correct, considering that other studies have tested this question before. Moreover, doing something for the first time is not per se a sign of quality. This phrase may be removed.**

This phrase has been removed.

Regarding comments of Reviewer 3:

This MS presents elegant experiments, demonstrating complex sleep dependent reconsolidation across destabilized memories. The most interesting aspect of the MS is the interactions across memories. Namely, learning a new task may temporarily impair performance of a previously learned task, but sleep might then improve the performance of both. This finding raises questions (which should be discussed) about the dynamics of complex learning, perhaps also in a sensory- motor context.

- 29. As I was reading the MS I kept wondering about possible ‘side effects’ of testing sessions on memories. Of course, testing sessions are designed to minimize such effects, but no details were presented in the Methods section about the testing procedure to allow judging it. Measurements always change the measured features, this is unavoidable. However, here this issue is of particular concern since conclusions depend on the assumption that testing does not induce further learning. The concern is that ‘second night’ is also a ‘first night’ after learning induced by the testing. Providing some more details about the testing and of possible ‘side effects’ of those sessions could help.**

Please see our response to point-2 above.

- 30. Figure numbers are not properly ordered: 1A then 2, then 1B... Please make figure numbers linear.**

We have revised the numbering of the figures

- 31. Figure 2b: classification performance at posttest-4 showed a significant improvement of 5.3 ± 1.3 percentage points after a second night of sleep. But it is not clear if this second improvement resulted in a higher level of performance compared to birds who experienced no interference? It is difficult to judge if the second improvement in Figure 2b could mirror a recovery from the slight deterioration during previous day, or (more interestingly) a real additional improvement step triggered by interference. This can be tested statistically across the two groups. Perhaps try testing if the final performance differ across the two groups?**

The performance improvement across the second night of sleep was 5.3 ± 1.3 for the early-interference condition and 1.1 ± 2.0 for the no-interference condition. This difference failed to reach significance ($t_{35} = 1.59$, $P = 0.12$). As a comparison, for experiment-2, the

average performance improvement across nights 2-4 was 4.9 ± 1.4 in the early-interference condition and 1.6 ± 1.1 in the no interference condition. This difference was considered to be marginally significant ($t_{23} = 2.18$, $P = 0.04$). Though the exact P value is less than 0.05, the comparison is only marginally significant ($P < 0.10$) when taking into account corrections for multiple comparisons.

32. I would suggest adding a discussion section focusing on possible implications for complex sensory-motor learning, such as song learning. Sleep appears to affect such learning as well, inducing oscillations that are structurally similar to those reported here. Perhaps authors could outline some preliminary framework for an extended framework based on multiple, hierarchical learning tasks, could link between the current paper and natural sensory-motor learning?

We have added comments to the discussion regarding how reconsolidation may be involved in the sensory and sensorimotor phases of song-learning.

REVIEWERS' COMMENTS:

Reviewer #1 (Remarks to the Author):

The authors have made massive and impressive changes to their manuscript. Most of my comments were adequately and extensively addressed to my satisfaction. I, therefore, stand by my recommendation to approve this manuscript for publication.

The only point that I still feel uncomfortable with concerned the answer to my first comment. The authors conducted several analyses to prove that the pattern of linear correlations between the measure of memory "loss" over a day and the subsequent sleep-related "gain" is similar across conditions. Additionally, they make the point that this "recuperation" over-sleep may not have been expected due to previous findings, such as the Walker et al., 2003 paper. Finally, they briefly present a model to explain their understanding of how reconsolidation and sleep may interact.

I do not dispute the authors' claims or their findings. My point, however, remains: the authors strongly suggest that reconsolidation may actually benefit memory traces, whereas what their findings show is more akin to recovery over sleep. In Experiment 2, the overall memory gain between test-1 and test-9 is higher for the No-Interference condition (although these differences between conditions are not significant). In Experiment 1, there are bigger improvements for the conditions including Task-B, but there doesn't seem to be any difference between the early- and late-intervention conditions in the overall benefit (test-1 vs. test-4), suggesting that reconsolidation did not benefit overall memory. The point is that parts of the discussion suggest that reconsolidation, in interaction with sleep, may be useful for memory improvement, whereas the manuscript does not show any indication of that, despite the intriguing and robust results showing destabilization and reconsolidation over sleep. Phrases such as "reconsolidation may help stabilize memory" or "classification memory was improved and more stable on the following day", both from the discussion, may mislead readers if they are not accompanied by the proper disclaimer. There is no indication for beneficial stabilization, just destabilization and then return to baseline. All memory improvements followed memory decline and the net gain was not significantly different than zero. As the authors mention, given the controversial literature regarding reconsolidation and sleep, their findings are exciting enough even without the claim of a beneficial overall effect of reconsolidation during sleep. I think the possibility of strengthening memory above its initial levels using consolidation is exciting and a discussion of this point with regard to the author's data would be a great addition to the manuscript.

Minor points:

a. In page 10, you mention that "whereas the no-interference condition showed non-significant changes [in experiment 1 & 2], classification performance in two interference conditions both improved significantly after sleep". The former claim is definitely true for experiment 1, but is it true for experiment 2? Even though there were no significant increases in memory in any single night, it seems the overall performance did increase over the five-day course.

b. In page 5, the authors write: "will learning a new pair of song stimuli in task-B after being tested on task-A produce similar performance impairments for task-A across waking retention on the second day as it does on the first day?" It is not clear to me which performance impairment for task-A during the first day the authors are referring to here.

Reviewer #2 (Remarks to the Author):

The authors have satisfactorily addressed all of my concerns and introduced appropriate changes in the manuscript. I have no further comments.

Reviewer #3 (Remarks to the Author):

authors have addressed all my comments. I have no further concerns.

Regarding comments of Reviewer 1:

The authors have made massive and impressive changes to their manuscript. Most of my comments were adequately and extensively addressed to my satisfaction. I, therefore, stand by my recommendation to approve this manuscript for publication.

1. The only point that I still feel uncomfortable with concerned the answer to my first comment. The authors conducted several analyses to prove that the pattern of linear correlations between the measure of memory "loss" over a day and the subsequent sleep-related "gain" is similar across conditions. Additionally, they make the point that this "recuperation" over-sleep may not have been expected due to previous findings, such as the Walker et al., 2003 paper. Finally, they briefly present a model to explain their understanding of how reconsolidation and sleep may interact. I do not dispute the authors' claims or their findings. My point, however, remains: the authors strongly suggest that reconsolidation may actually benefit memory traces, whereas what their findings show is more akin to recovery over sleep. In Experiment 2, the overall memory gain between test-1 and test-9 is higher for the No-Interference condition (although these differences between conditions are not significant). In Experiment 1, there are bigger improvements for the conditions including Task-B, but there doesn't seem to be any difference between the early- and late-intervention conditions in the overall benefit (test-1 vs. test-4), suggesting that reconsolidation did not benefit overall memory. The point is that parts of the discussion suggest that reconsolidation, in interaction with sleep, may be useful for memory improvement, whereas the manuscript does not show any indication of that, despite the intriguing and robust results showing destabilization and reconsolidation over sleep. Phrases such as "reconsolidation may help stabilize memory" or "classification memory was improved and more stable on the following day", both from the discussion, may mislead readers if they are not accompanied by the proper disclaimer. There is no indication for beneficial stabilization, just destabilization and then return to baseline. All memory improvements followed memory decline and the net gain was not significantly different than zero. As the authors mention, given the controversial literature regarding reconsolidation and sleep, their findings are exciting enough even without the claim of a beneficial overall effect of reconsolidation during sleep. I think the possibility of strengthening memory above its initial levels using consolidation is exciting and a discussion of this point with regard to the author's data would be a great addition to the manuscript.

We have removed the phrase "classification memory was improved and more stable on the following day" from the 1st paragraph of the discussion as that could be misleading because the improvement was relative to performance levels in the evening, which were non-significantly (early-interference) or significantly (late-interference) lower than the earlier test at noon. Thus, the reconsolidation could represent recovery rather than strengthening above the original levels. We have kept the phrase "reconsolidation may help to stabilize the memory..." from the 3rd-to-last paragraph in the discussion. We believe the data do support the claim that reconsolidation helped to stabilize memory. Indeed, for the early-interference condition in experiment-2, the interference tasks did not impair task-A memory when they were learned in the morning prior to the task-A retests. If sleep had not reconsolidated (i.e., re-stabilized) the memory, then the task-A

memory would still be destabilized from the prior task-A test, which would allow the interference tasks to subsequently impair task-A performance as observed in the late-interference conditions. We have additionally added a sentence to the 4th-to-last paragraph raising the possibility of using consolidation and memory interactions to optimize memory strengthening.

Minor points:

- 2. In page 10, you mention that "whereas the no-interference condition showed non-significant changes [in experiment 1 & 2], classification performance in two interference conditions both improved significantly after sleep". The former claim is definitely true for experiment 1, but is it true for experiment 2? Even though there were no significant increases in memory in any single night, it seems the overall performance did increase over the five-day course.**

We believe the claim is correct for both experiments. While it is true that the no-interference condition in experiment-2 showed a significant improvement from test-1 to test-9 across the 5 days, the performance change over sleep was not significant for any night and was nearly identical to the incremental performance increases that occurred across the waking retention periods. Thus, the no-interference condition did not show sleep-dependent gains, but the early-interference and late-interference improvements only occurred after sleep.

- 3. In page 5, the authors write: "will learning a new pair of song stimuli in task-B after being tested on task-A produce similar performance impairments for task-A across waking retention on the second day as it does on the first day?" It is not clear to me which performance impairment for task-A during the first day the authors are referring to here.**

The "first day" was referring to the task-A performance impairments that were observed in our previous work when task-A and task-B were learned on the same day, in contrast to the current experiment where task-B was learned on the second day. We have removed the phrase "as it does on the first day" to avoid this ambiguity.